# Reactive oxygen species prevent lysosome coalescence during PIKfyve inhibition

**Golam T. Saffi**[1,2,3], **Evan Tang**[2], **Sami Mamand**[1,4], **Subothan Inpanathan**[1,2], **Aaron Fountain**[1,2], **Leonardo Salmena**[3,5], **Roberto J. Botelho**[1,2]*

**1** Molecular Science Graduate Program, Ryerson University, Toronto, Ontario, Canada, **2** Department of Chemistry and Biology, Ryerson University, Toronto, Ontario, Canada, **3** Department of Pharmacology & Toxicology, University of Toronto, Toronto, Ontario, Canada, **4** Polytechnic Research Center, Erbil Polytechnic University, Kurdistan Regional Government, Erbil, Kurdistan, **5** Princess Margaret Cancer Centre, University Health Network, Toronto, Ontario, Canada

* rbotelho@ryerson.ca

**Data Availability Statement:** Data for Western blots is attached as Supplemental Information S22 Western Blots. Other data (figures and spreadsheets) are available publicly and without

## Abstract

Lysosomes are terminal, degradative organelles of the endosomal pathway that undergo repeated fusion-fission cycles with themselves, endosomes, phagosomes, and autophagosomes. Lysosome number and size depends on balanced fusion and fission rates. Thus, conditions that favour fusion over fission can reduce lysosome numbers while enlarging their size. Conversely, favouring fission over fusion may cause lysosome fragmentation and increase their numbers. PIKfyve is a phosphoinositide kinase that generates phosphatidylinositol-3,5-bisphosphate to modulate lysosomal functions. PIKfyve inhibition causes an increase in lysosome size and reduction in lysosome number, consistent with lysosome coalescence. This is thought to proceed through reduced lysosome reformation and/or fission after fusion with endosomes or other lysosomes. Previously, we observed that photo-damage during live-cell imaging prevented lysosome coalescence during PIKfyve inhibition. Thus, we postulated that lysosome fusion and/or fission dynamics are affected by reactive oxygen species (ROS). Here, we show that ROS generated by various independent mechanisms all impaired lysosome coalescence during PIKfyve inhibition and promoted lysosome fragmentation during PIKfyve re-activation. However, depending on the ROS species or mode of production, lysosome dynamics were affected distinctly. $H_2O_2$ impaired lysosome motility and reduced lysosome fusion with phagosomes, suggesting that $H_2O_2$ reduces lysosome fusogenecity. In comparison, inhibitors of oxidative phosphorylation, thiol groups, glutathione, or thioredoxin, did not impair lysosome motility but instead promoted clearance of actin puncta on lysosomes formed during PIKfyve inhibition. Additionally, actin depolymerizing agents prevented lysosome coalescence during PIKfyve inhibition. Thus, we discovered that ROS can generally prevent lysosome coalescence during PIKfyve inhibition using distinct mechanisms depending on the type of ROS.

## Introduction

Lysosomes are typically defined as terminal organelles with an acidic and degradative lumen that digest macromolecules received through endocytosis, phagocytosis and autophagy [1–3]. In reality, lysosomes are part of an endomembrane spectrum formed through heterotypic and

restrictions in FigShare public repository of data with the DOI 10.6084/m9.figshare.16663720.

**Funding:** This work was funded by grants awarded to RJB from the Natural Sciences and Engineering Council of Canada (Grant numbers RGPIN-2015-06489, 441595-2013, RGPIN-2020-04343; https://www.nserc-crsng.gc.ca) the Canada Research Chairs Program (https://www.chairs-chaires.gc.ca), Canada Foundation for Innovation (32957; https://www.innovation.ca/) an Early Researcher Award from the Government of Ontario (ER13-09-042; https://www.ontario.ca/page/early-researcher-awards), and contributions from Ryerson University. The funders had no role in the study design, data collection and analysis, decision to publish, or preparation of the manuscript.

**Competing interests:** The authors have declared that no competing interests exist.

**Abbreviations:** BAPTA-AM, 1,2-Bis(2-aminophenoxy)ethane-N,N,N′,N′-tetraacetic acid tetrakis-acetoxymethyl ester; CDNB, 1-chloro-2,4,-dinitrobenzene; DMEM, Dulbecco's Modified Eagle Medium; FBS, fetal bovine serum; FYCO1, FYVE and Coiled-Coil Domain Autophagy Adaptor-1; HOPS, Homotypic Fusion Protein Sorting; HPF, hydroxylphenyl fluorescein; LAMP1, lysosomal membrane protein-1; LC3, Light Chain-3; LLMeO, L-leucyl-L-leucine methyl ester; MCB, monochlorobimane; MCOLN1, Mucolipin-1; NAC, N-acetyl-L-cysteine; NBT, Nitroblue tetrazolium; PFA, paraformaldehyde; PIKfyve, Phosphoinositide Kinase, FYVE-type Zinc Finger Containing; PLEKHM2, Pleckstrin-Homology and RUN domain containing M2; PtdIns(3)P, phosphatidylinositol-3-phosphate; PtdIns(3,5)$P_2$, phosphatidylinositol-3,5-bisphosphate; PtdIns(4,5)$P_2$, phosphatidylinositol-4,5-bisphosphate; PtdInsP, phosphoinositide; RILP, Rab-Interacting Lysosomal Protein; ROS, reactive oxygen species; RPE cells, retinal pigment epithelium cells.

homotypic fusion between late endosomes that enclose cargo for degradation, terminal lysosomes, which are non-acidic, hydrolase-dormant storage organelles, and endolysosomes, hybrids formed when late endosomes and terminal lysosomes fuse together [4–7]; we use the term *lysosome* to refer to this spectrum. Importantly, fusion and content exchange along the lysosomal spectrum proceeds through two major routes. First, lysosomes can fuse with a target organelle resulting in complete merger of the two compartments. Alternatively, lysosomes can exchange content with another target organelle through "kiss-and-run"; in this process, a transient fusion between two organelles generates an aqueous pore to exchange content and is followed by fission to prevent amalgamation of the two compartments [6, 8–10].

Delivery of cargo to lysosomes is an incessant process that depends on cargo sorting, membrane targeting, and the fusion machinery, which are governed by, among others, the lysosomal GTPases, Rab7 and Arl8b [11, 12]. These GTPases modulate the movement of lysosomes along microtubule tracks through their effectors; Rab7 uses Rab-Interacting Lysosomal Protein (RILP) and FYVE and Coiled-Coil Domain Autophagy Adaptor-1 (FYCO1) to engage dynein and kinesins, thus moving lysosomes towards the cell centre and periphery, respectively [13, 14]. In comparison, Arl8b uses Pleckstrin-Homology and RUN domain containing M2 (PLEKHM2; or SKIP) protein to engage kinesin to promote lysosome positioning to the cell periphery [15]. When lysosomes contact other lysosomes/late endosomes, this engages tether complexes like Homotypic Fusion Protein Sorting (HOPS) complex, also modulated by Rab7 and Arl8b, and eventually undergo fusion [12, 16, 17]. Lysosome fusion and fission dynamics is also modulated by intralysosomal $Ca^{2+}$ release via Mucolipin-1 (MCOLN1) and P2X4 channels [18, 19].

Despite the incessant delivery of content to lysosomes through fusion, cells maintain lysosome number and size, suggesting that exit of cargo from lysosomes by fission is also relentless. Yet, much less is known about lysosome fission, which may proceed through vesiculation, tubulation, and splitting [10]. Lysosome fission mechanisms may include classical coat and fission machinery such as clathrin and dynamin and actin complexes [10, 20–24]. Coordination of these fission machines is poorly understood but likely involves MCOLN1-$Ca^{2+}$ release [19, 25, 26]. Additionally, phosphoinositides (PtdInsPs) play a key role in lysosome fission dynamics including modulation of vesiculation versus tubulation [10]. Amongst these, lysosome fission-fusion cycles are coordinated by the Phosphoinositide Kinase, FYVE-type Zinc Finger Containing (PIKfyve) lipid kinase that synthesizes phosphatidylinositol-3,5-bisphosphate [PtdIns(3,5)$P_2$] and directly or indirectly, phosphatidylinositol-5-phosphate [27, 28]. Pharmacological or genetic disruption of PIKfyve and partner proteins like Vac14 and the Fig 4 phosphatase cause enlarged lysosomes, partly by impairing fission and reformation of terminal lysosomes [4, 27, 29–31]. The result is lysosome coalescence, enlarging lysosomes while reducing their numbers [4, 30]. It remains unclear how PIKfyve controls lysosome fission but may involve control of actin-assembly on lysosomes and fission proteins [19, 21, 26, 32].

During our studies with acute PIKfyve inhibition, we observed that imaging by spinning disc confocal microscopy at high frequency inhibited lysosome enlargement caused by PIKfyve inhibition [30]. We speculated that this resulted from photo-generated reactive oxygen species (ROS), which can include superoxide anions ($O_2^-$), hydrogen peroxide ($H_2O_2$), and hydroxyl radicals ($OH^-$) [33–35]. ROS species are also formed as part of normal aerobic metabolism and can actually be produced as signaling intermediates to modulate cell proliferation and the inflammatory response [36, 37]. Yet, overt ROS production is detrimental, damaging proteins, lipids, and DNA. In fact, conditions that increase oxidative load on cells, such as inhibition of thioredoxin, can disrupt lysosome and autophagy systems via membrane damage [38, 39]. Thus, cells have evolved multiple systems to quench ROS levels including $O_2^-$ dismutase, catalase, glutathione, and thioredoxin [37, 40].

In this study, we sought to understand if other modes of ROS generation could abate lysosome coalescence during PIKfyve inhibition and to better define the mechanisms of action. Strikingly, we found that different sources of ROS reduced lysosome coalescence during PIKfyve inhibition and promoted lysosome fragmentation upon PIKfyve reactivation. Interestingly, these distinct ROS hindered lysosome coalescence differently. $H_2O_2$ prevented lysosome coalescence by impairing lysosome motility and blunting lysosome fusogenecity. In comparison, oxidative decoupling of the mitochondria with rotenone and inhibitors of glutathione, thiol groups, or thioredoxin counteracted lysosome coalescence by releasing actin clusters that accumulated on lysosomes during PIKfyve impairment.

## Results

### Stimulation of ROS suppresses lysosome enlargement during acute PIKfyve inhibition

We previously observed that extended laser excitation by spinning disc confocal fluorescence microscopy arrested lysosome enlargement during acute PIKfyve suppression [30]. We speculated that this arrest may be due to ROS production caused by light energy [41]; in fact, RAW cells that were more frequently exposed to laser light displayed higher levels of nitroblue tetrazolium (NBT) staining, a ROS indicator (S1A Fig). This led us to hypothesize that other mechanisms of ROS generation could impair lysosome enlargement during acute inhibition of PIKfyve. To test this, we exposed cells to a variety of ROS inducers: $H_2O_2$, rotenone, which decouples the mitochondrial electron chain, monochlorobimane (MCB), a glutathione S-transferase inhibitor, or to a thiol inhibitor, 1-chloro-2,4,-dinitrobenzene (CDNB) [35, 42–44]. We first demonstrated that these manipulations enhanced ROS levels by using CellROX Green, a redox sensitive dye whose fluorescence is proportional to ROS levels (Fig 1A and 1B). Additionally, ROS cause MCB to form fluorescent MCB-glutathione adducts; we observed 7x more MCB-glutathione adducts relative to vehicle (Fig 1C and 1D). To better define the type(s) of ROS generated by these treatments, we used fluorescent detectors for $O_2^-$ (ROS-ID), mitochondrial $O_2^-$ (MitoSox), $OH^./$peroxynitrite (HPF), and singlet $O_2^.$ (si-DMA). We found that $H_2O_2$ was the most promiscuous agent generating all species except detectable levels of singlet $O_2^.$ (Fig 1E–1H). In turn, rotenone generated mitochondrial $O_2^-$ and singlet $O_2^.$ as detected by MitoSox and si-DMA, respectively (Fig 1E2–1H), while CDNB favoured production of singlet $O_2^.$ (Fig 1E–1H). MCB did not elicit detectable changes in these probes, though ROS were detected when using CellRox and GSH-MCB (Fig 1D). Despite the production of various ROS, none of the treatments caused appreciable cell death over 2 h of exposure, as detected by propidium staining (Fig 1I).

Remarkably, we then observed that all ROS inducers arrested lysosome enlargement in cells treated with apilimod, a potent and selective PIKfyve blocker (Fig 2). More specifically, RAW cells treated with apilimod alone suffered an increase in the size of individual lysosomes (Fig 2B) and a decrease in lysosome number (Fig 2C), indicating that lysosomes coalesced. As we documented before, the total cellular volume of the lysosome population was unchanged between resting and apilimod-treated cells (Fig 2D). In comparison, co-exposure of cells with apilimod to either $H_2O_2$, rotenone, MCB or CDNB prevented lysosome enlargement and reduction in lysosome number (Fig 2A–2D). Moreover, we used auranofin as thioredoxin reductase inhibitor and to complement the general thiol inhibitor, CDNB, and observed that it too prevented apilimod-induced lysosome coalescence (Fig 2A–2D). To test whether lower $H_2O_2$ levels (100 μM) could also block lysosome coalescence, we used lower apilimod concentrations (1 or 5 nM). We still observed lysosome coalescence at these lower concentrations of apilimod and this was prevented in cells that were co-exposed to 100 μM $H_2O_2$ (S2 Fig). No

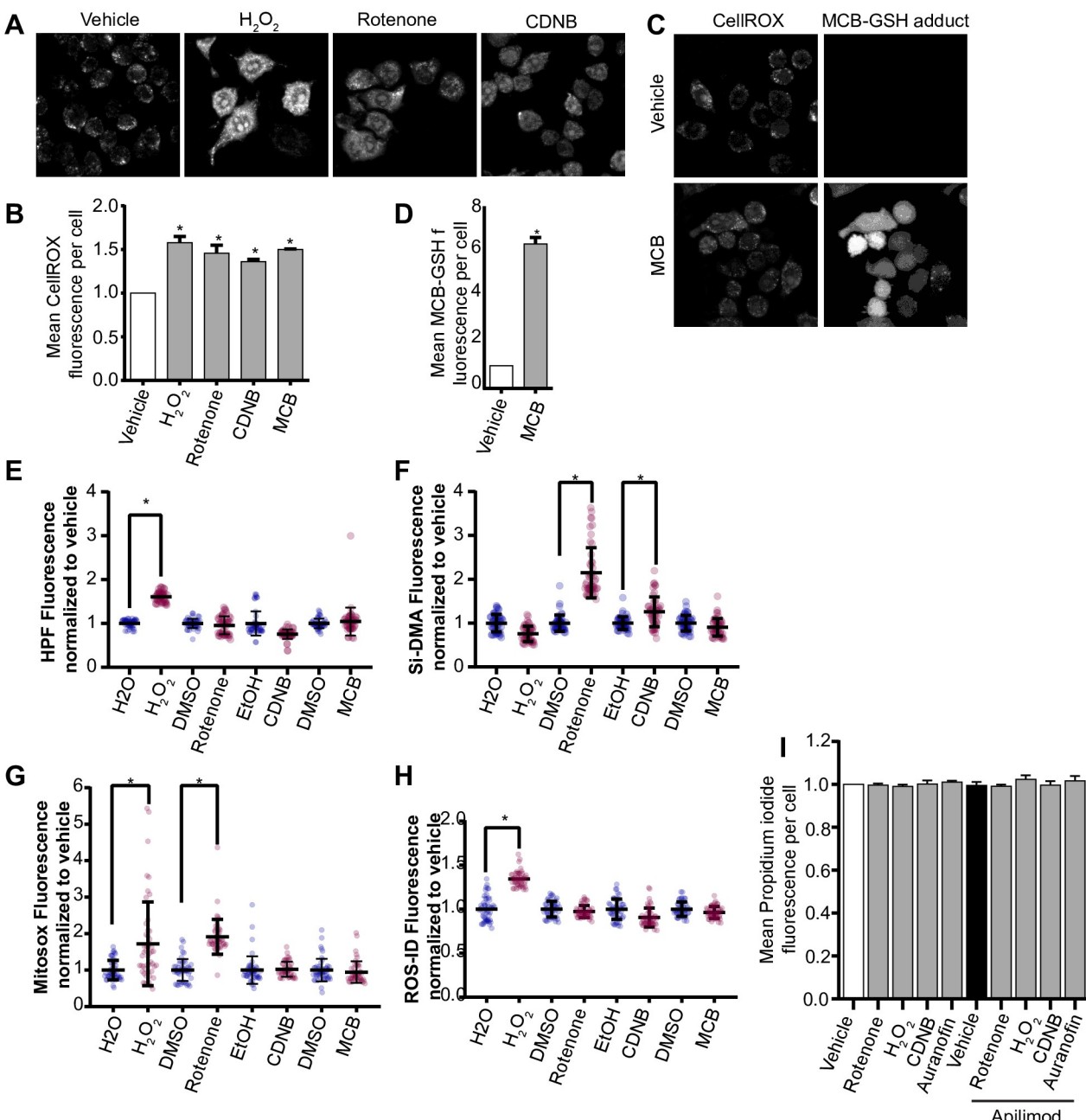

**Fig 1. Different ROS inducers produce different ROS in RAW macrophages.** (A) RAW cells were exposed to vehicle, or to one of the following ROS inducers– 1 mM $H_2O_2$ 40 min, 1 μM rotenone 60 min, 10 μM CDNB 30 min, or 5 μM MCB 30 min. Cells were then stained with CellROX Green to detect and quantify the levels of ROS formed during these treatments. Fluorescence micrographs represent single z-focal plane images from spinning disc confocal microscopy. Scale bar = 20 μm. (B) Quantification of CellROX Green fluorescence intensity. C) MCB-GSH adduct was also detected during vehicle or MCB treatment. Fluorescence micrographs represent single z-focal plane images from spinning disc confocal microscopy. Scale bar = 20 μm. (D) Quantification of MCB-GSH fluorescence intensity. E-H: Quantification of ROS-specific probes, where HPF detects hydroxyl and perinitrite (E), Si-DMA detects singlet oxygen (F), Mitosox detects mitochondrial superoxide (G), and ROS-ID detects cytoplasmic superoxide (H). For each ROS probe (burgundy dots), fluorescence was normalized against the respective vehicle control (indigo dots). (I) RAW cells were exposed to vehicle or apilimod, and/or to one of the following ROS inducers– 1 mM $H_2O_2$ 40 min, 1 μM rotenone 60 min, 10 μM CDNB 30 min, or 5 μM MCB 30 min. Cells were then stained, analyzed for propidium iodide intensity, and normalized to vehicle-control to measure cell viability. For all graphs, data are represented as mean ± S.E.M. from three independent experiments with 40–50 cells assessed per treatment per experiment. One-way ANOVA and Tukey's *post-hoc* test were used for (B, E-I), and an unpaired Student's t-test performed for (D). * indicates $p < 0.05$.

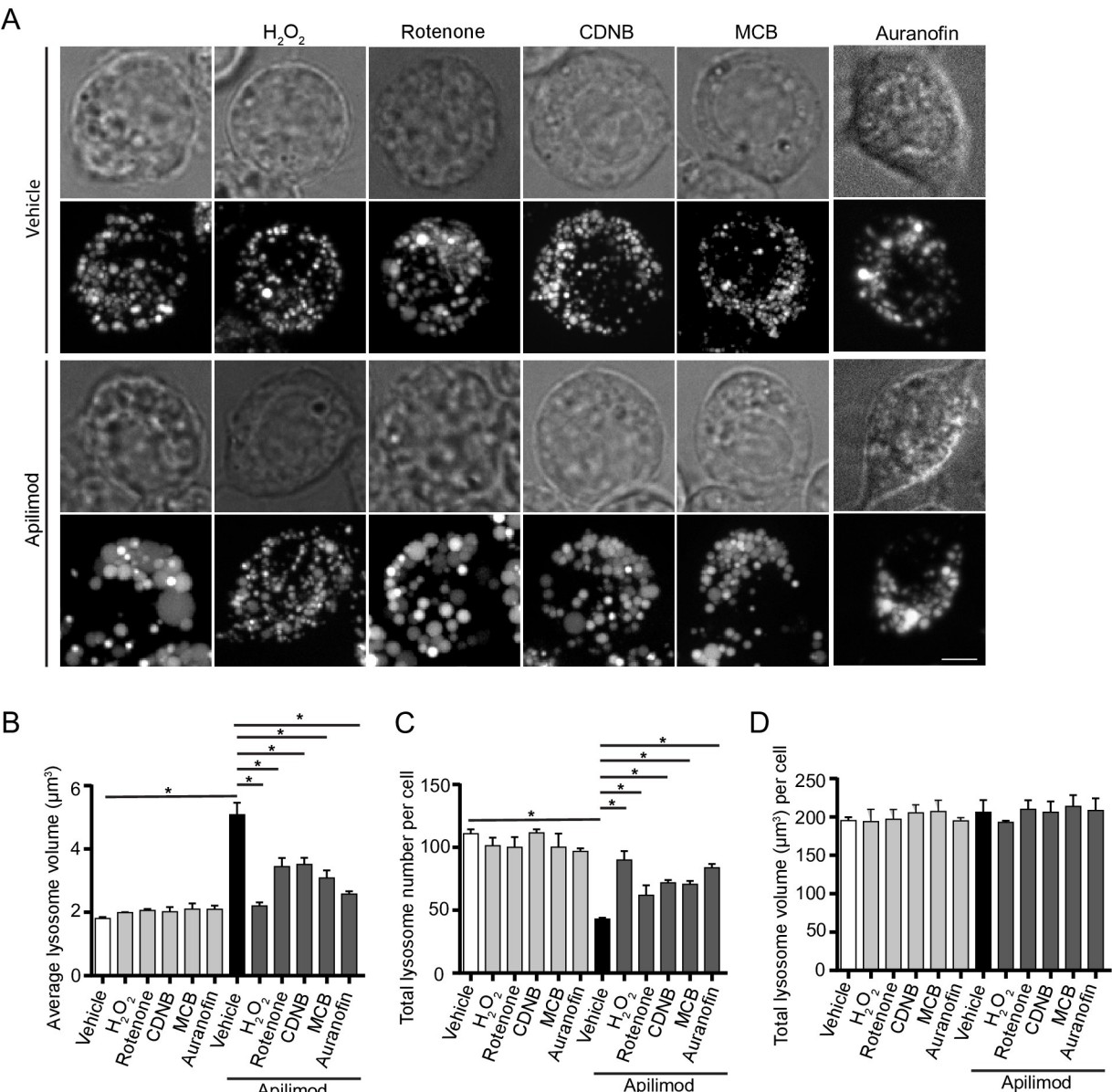

**Fig 2. ROS agonists prevent lysosome enlargement during acute PIKfyve suppression.** *A*: RAW cells pre-labelled with Lucifer yellow and exposed to vehicle or 20 nM apilimod for 40 min. These conditions were then supplemented with additional vehicle or 1 mM $H_2O_2$ for 40 min, 1 μM rotenone for 60 min, 10 μM CDNB for 30 min, 5 μM MCB for 30 min or auranofin 10 μM for 120 min. Fluorescence micrographs are represented as z-projections of 45–55 z-plane images obtained by spinning disc microscopy. Scale bar: 5 μm. *B-D*: Quantification of individual lysosome volume (B), lysosome number per cell (C), and total lysosome volume per cell (D). Data represent mean ± S.E.M. from three independent experiments, with 25–30 cells assessed per treatment condition per experiment. One-way ANOVA and Tukey's *post-hoc* test was used, where * indicates statistical significance between indicated conditions ($p < 0.05$).

significant changes to lysosome number, size of individual lysosomes and total lysosome volume were observed when ROS agonists were used alone (Fig 2A–2D). The fact that ROS alone did not appear to further reduce lysosome size and increase lysosome number may reflect some physical restriction to the smallest lysosome size; for example, osmotic pressure may prevent a further reduction in the size of basal lysosomes.

To provide evidence that ROS were the active agents that blocked lysosome coalescence during apilimod-treatment, we employed N-acetyl-L-cysteine (NAC) as an anti-oxidant during rotenone co-administration [45]. Indeed, cells co-exposed with apilimod, rotenone and NAC displayed larger lysosomes than cells co-treated with apilimod and rotenone (Fig 3), indicating that ROS are the active agents arresting lysosome enlargement during PtdIns(3,5)P$_2$ depletion.

To ensure that these observations were not specific to murine RAW macrophages, we assessed apilimod-induced lysosome coalescence in human-derived RPE and HeLa cells co-exposed to H$_2$O$_2$ or CDNB. As with RAW cells, while apilimod-alone induced lysosome coalescence in HeLa and RPE cells, counterpart cells co-administered apilimod and H$_2$O$_2$ or CDNB resisted lysosome enlargement and reduction in lysosome number (S3 Fig). As before, no changes were observed under any treatment to the total lysosomal volume within these cell types (S3D and S3H Fig). Overall, our observations suggest that generation of ROS via distinct mechanisms can impair lysosome coalescence caused by PIKfyve inhibition in several cell types.

## ROS help reverse lysosome fragmentation during PIKfyve reactivation

Removal of apilimod elicited reversal of lysosome coalescence, re-establishing lysosome size and number after >3 h post drug removal [30]. To test if ROS exposure could help reverse

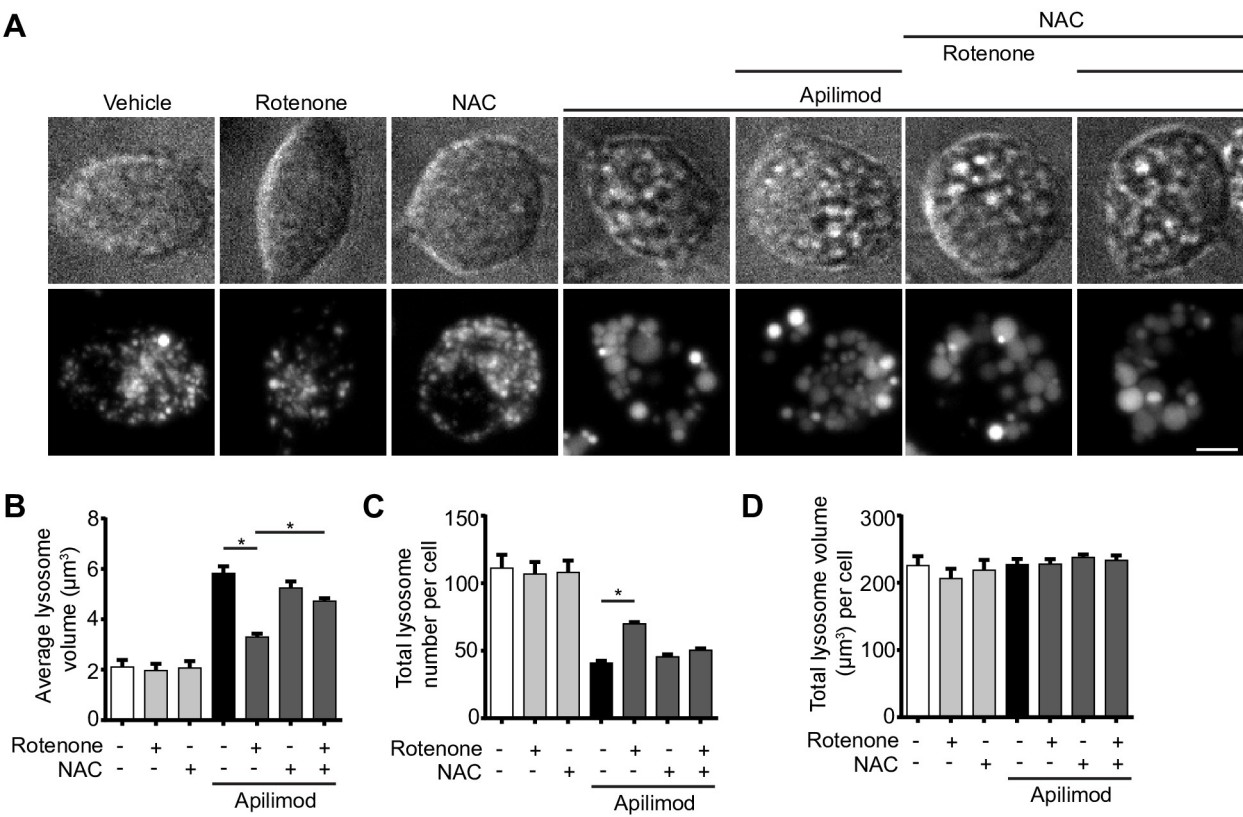

**Fig 3. ROS scavengers permit lysosome coalescence during acute PIKfyve suppression.** *A*: RAW cells pre-labelled with Lucifer yellow and exposed to vehicle, or 0.5 μM rotenone 60 min, or 10 mM N-acetyl-L-cysteine (NAC) 120 min alone, or in presence of 20 nM apilimod for the last 40 min. Fluorescence micrographs are represented as z-projections of 45–55 z-plane images obtained by spinning disc microscopy. Scale bar: 5 μm. *B-D*: Quantification of individual lysosome volume (B), lysosome number per cell (C), and total lysosome volume per cell (D). Data represent mean ± S.E.M. from three independent experiments, with 25–30 cells assessed per treatment condition per experiment. One-way ANOVA and Tukey's *post-hoc* test was used, where * indicates statistical significance between indicated conditions ($p < 0.05$).

lysosome coalescence, we treated RAW cells with apilimod for 1 h and then incubated cells with fresh, drug-free medium to reactivate PIKfyve with or without $H_2O_2$, rotenone, CDNB, or MCB during this wash duration. As before, apilimod increased lysosome size and decreased lysosome number, while chasing cells for 2 h after drug removal reversed this phenotype partly; longer incubation ultimately reverses lysosome enlargement completely [30]. Exposure to any of the ROS agents during the apilimod-free chase caused individual lysosomes to become smaller in size and more numerous relative to apilimod-wash only (Fig 4A–4D). Overall, ROS prevent lysosome coalescence induced by PIKfyve inhibition and help to reverse this upon PIKfyve reactivation. We next examined levels of PtdIns(3,5)P$_2$, lysosome membrane damage, lysosome motility, fusion and fission events in order to better understand the mechanisms that enable ROS to prevent lysosome coalescence in the absence of PIKfyve activity.

## ROS stimulation arrests apilimod induced lysosome enlargement without neutralizing apilimod or stimulating PtdIns(3,5)P2 synthesis

To understand the effect of ROS on apilimod-mediated lysosome enlargement, we first considered the trivial possibility that higher ROS load within cells may degrade the structural integrity of apilimod, relieving the acute PIKfyve suppression, and thus preventing lysosome coalescence. To test this, we co-incubated apilimod with $H_2O_2$ in complete medium *in vitro* for 40 min. Following this incubation, we added catalase to decompose $H_2O_2$ and then transferred the reaction mixture onto RAW macrophages to observe if apilimod was still able to induce lysosome enlargement. We found that apilimod pre-exposed to $H_2O_2$ was still able to increase lysosome size and decrease lysosome number similarly to an aliquot of naïve apilimod (Fig 5A–5D), suggesting that $H_2O_2$ did not degrade apilimod. Moreover, while $H_2O_2$ arrested apilimod-mediated lysosome enlargement, the co-addition of catalase to apilimod and $H_2O_2$ permitted lysosome enlargement, further suggesting that $H_2O_2$ is a direct suppressor of lysosome coalescence in PIKfyve-inhibited cells (Fig 5A–5D). Therefore, we provide additional evidence that ROS rescue lysosome coalescence during acute PIKfyve inhibition.

We next examined if ROS rescue lysosome coalescence during PIKfyve inhibition by increasing the levels of PtdIns(3,5)P$_2$ in cells. In part, this may occur because ROS species reversibly oxidize catalytic cysteine residues on protein and lipid phosphatases, abating their activity [37, 46, 47]. Therefore, augmented ROS levels may inhibit the Fig 4 lipid phosphatase, counteracting PIKfyve inhibition with apilimod and boosting PtdIns(3,5)P$_2$ levels [31]. This putative PtdIns(3,5)P$_2$ elevation may then be sufficient to prevent lysosome coalescence in cells exposed to apilimod and ROS. To test this hypothesis, we measured PtdInsP levels in cells treated with $H_2O_2$ or rotenone with and without apilimod by labelling cells with $^3$H-*myo*-inositol and using HPLC-coupled flow scintillation [48]. However, we observed a similar drop of about 80% in Ptdns(3,5)P$_2$ in cells treated with apilimod with or without ROS agents (Fig 5E and 5F), suggesting that ROS stimulation does not significantly elevate PtdIns(3,5)P$_2$ levels. In addition, inhibition of PIKfyve typically causes an increase in PtdIns(3)P levels (Fig 5E and 5F). While rotenone had no effect on this increase, $H_2O_2$ appeared to prevent this spike in PtdIns(3)P levels during apilimod treatment (Fig 5E). The significance of this change is not clear to us but given that rotenone still increased PtdIns(3)P and prevented lysosome coalescence, it is not likely to explain our observations. Lastly, PtdIns(4,5)P$_2$ has been linked to lysosome reformation and dynamics after autophagy [23]. However, we did not observe changes to this lipid in cells treated with apilimod and/or H2O2 or rotenone (Fig 5G and 5H). Overall, ROS prevents lysosome coalescence during PIKfyve inhibition via a mechanism that is independent of PtdIns(3,5)P$_2$ levels.

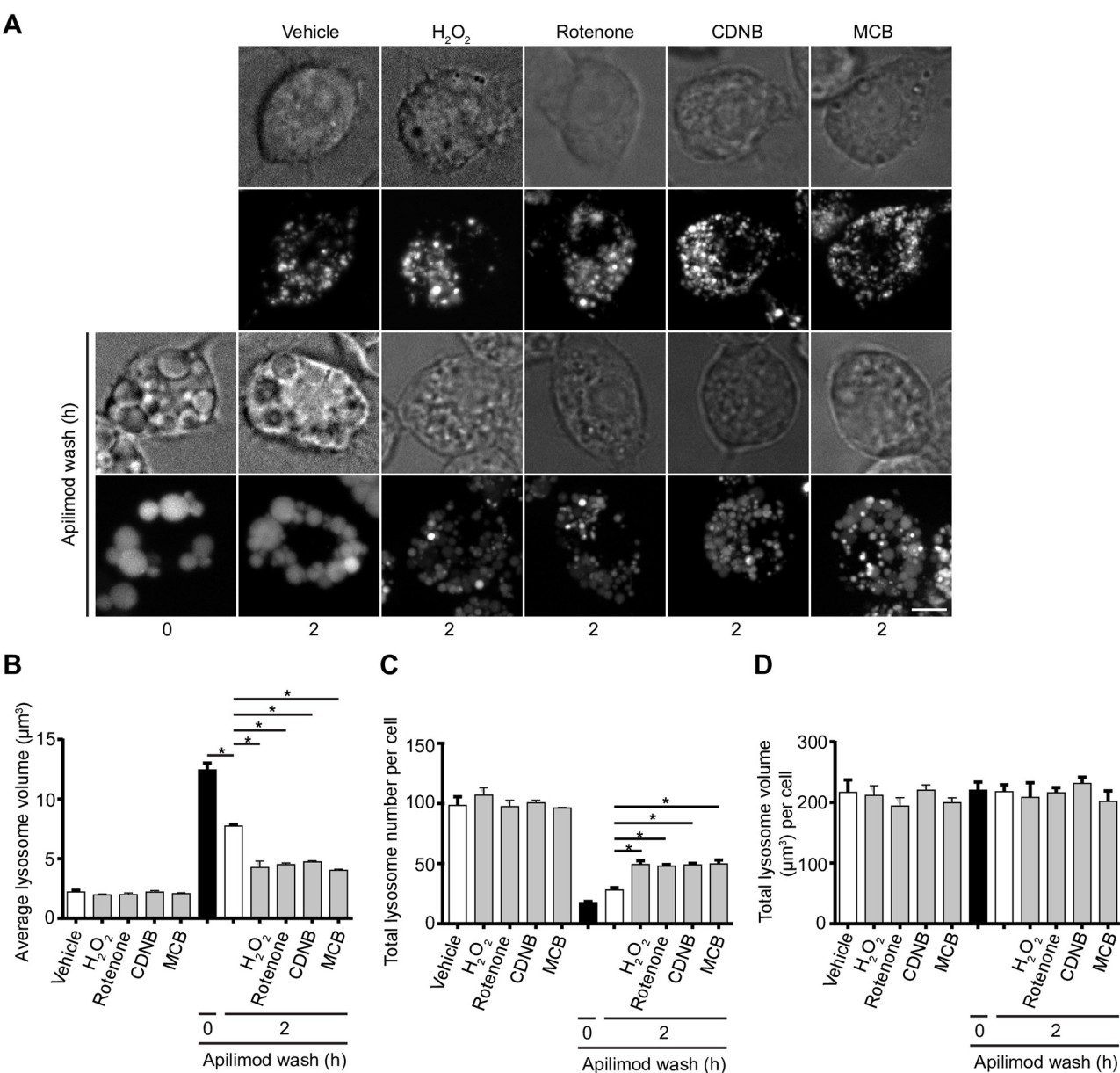

**Fig 4. ROS accelerate recovery of lysosome size and number upon PIKfyve reactivation.** (A) Top two rows: RAW cells pre-labelled with Lucifer yellow were exposed to either vehicle, 1 mM $H_2O_2$ 40 min, 1 μM rotenone 60 min, 10 μM CDNB 30 min, or 5 μM MCB 30 min. Bottom two rows: alternatively, RAW cells were first treated with 20 nM apilimod for 60 min (0 h), followed by apilimod removal and replenishment with complete media for 2 h in the presence of vehicle, $H_2O_2$, rotenone, CDNB, or MCB at previously indicated concentrations. Fluorescence micrographs are spinning disc microscopy images with 45–55 z-planes represented as z-projections. Scale bar: 5 μm. (B-D) Quantification of individual lysosome volume (B), lysosome number per cell (C), and total lysosome volume per cell (D). Data are represented as mean ± s.e.m. from three independent experiments, with 25–30 cell assessed per treatment condition per experiment. One-way ANOVA and Tukey's *post-hoc* test used for B-D, where * indicates statistically significant difference between control conditions ($P<0.05$).

## ROS alter the microtubule system

Since PtdIns(3,5)$P_2$ levels do not illuminate how ROS prevent lysosome coalescence during PIKfyve inhibition, we assessed other processes that affect lysosome dynamics. First, we examined whether ROS altered the microtubule system given its role in facilitating homotypic and heterotypic lysosome fusion. In fact, we previously showed that disruption

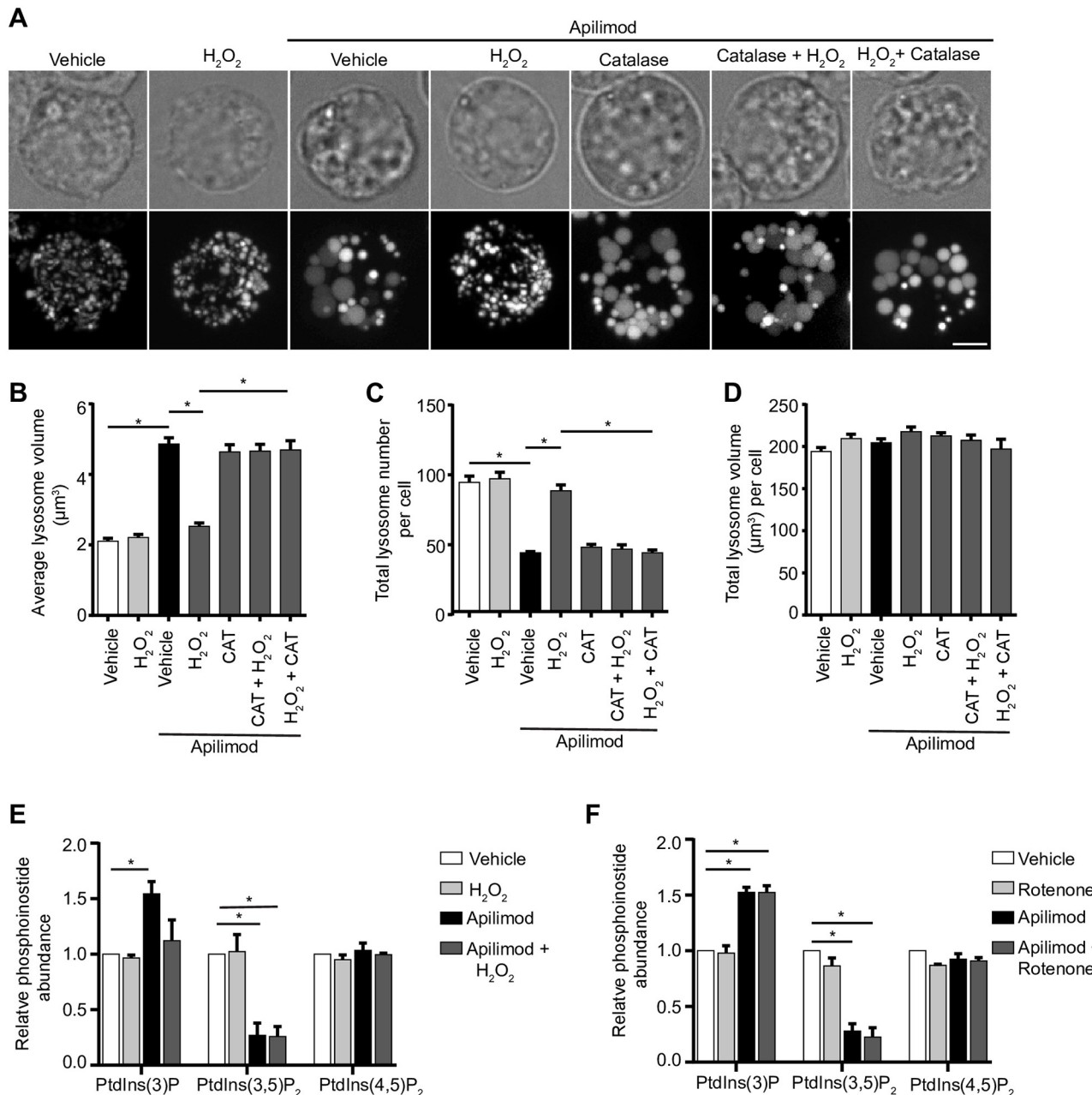

**Fig 5. Apilimod integrity and PtdIns(3,5)P$_2$ levels are not altered by ROS.** (A) RAW cells pre-labelled with Lucifer yellow. Following reactions were performed in complete media in vitro for designated time, prior to adding to cells for an additional 40 min: vehicle; 1 mM H$_2$O$_2$ 40 min; 20 nM apilimod 40 min; 20 nM apilimod preincubated with 1 mM H$_2$O$_2$ for 40 min; 20 nM apilimod preincubated with 0.5 mg/L catalase for 60 min; 1 mM H$_2$O$_2$ exposed to 0.5 mg/L catalase for 60 min to neutralize H$_2$O$_2$, followed by 20 nM apilimod 40 min; or 20 nM apilimod exposed to 1 mM H$_2$O$_2$ for 40 min to test whether H$_2$O$_2$ degraded apilimod, followed by 0.5 mg/L catalase for 60 min to degrade H$_2$O$_2$. Fluorescence micrographs are spinning disc microscopy images with 45–55 z-planes represented as z-projections. Scale bar: 5 μm. (B-D) Quantification of individual lysosome volume (B), lysosome number per cell (C), and total lysosome volume per cell (D). AP (apilimod), CAT (catalase). Data are shown as mean ± s.e.m. from three independent experiments, with 25–30 cell assessed per treatment condition per experiment. One-way ANOVA and Tukey's *post-hoc* test used for B-D; * indicates statistical difference against control condition (*P*<0.05). (E-F) $^3$H-*myo*-inositol incorporation followed by HPLC-coupled flow scintillation used to determine PtdIns(3)P, PtdIns(3,5)P$_2$ and PtdIns(4,5)P$_2$ levels from RAW cells exposed to vehicle alone, or 1 mM H$_2$O$_2$ 40 min (E), or 1 μM rotenone 60 min (F), in presence or absence of 20 nM apilimod. Data represent ± s.d. from three independent experiments. One-way ANOVA and Tukey's *post-hoc* test used for E-F; * indicates statistical difference against control condition (*P*<0.05).

of the microtubule system and microtubule motor activity blocked lysosome coalescence during PIKfyve inhibition [30]. We inspected the microtubule system in RAW macrophages (Fig 6A–6D) and RPE cells (Fig 6E–6H) exposed to the ROS agents by immunofluorescence staining against α-tubulin. We observed that the ROS agonists altered the microtubule system, but in distinct ways. Relative to untreated RAW macrophages or RPE cells, qualitative analysis of immunofluorescence images suggest that $H_2O_2$ makes microtubules more stable and extended, whereas increasing concentrations of rotenone, CDNB and MCB seemed to depolymerize microtubules, resulting in shorter microtubules and diffused staining (Fig 6). As a proxy to quantify changes to the microtubule morphology, we employed and validated the use of ImageJ "skeleton" plugin to extract different parameters of microtubule structure; these included filament junctions, branching, branch length, and patch area (S4 Fig). It is important to state that these are proxies rather than absolute descriptors of microtubule morphology. Using these measures, we were able to quantitatively show that all four ROS types altered the microtubule system with distinct effects. Briefly, $H_2O_2$ increased the number of microtubule junctions per cell and branch length in both RAW and RPE cells (Fig 6B, 6D, 6F and 6H). In comparison, rotenone, CDNB and MCB decreased the number of microtubule junctions and branches per cell significantly in RAW cells and increased the patch area in RPE cells (Fig 6B, 6C and 6I). These observations indicate that type of ROS and/or the site of ROS synthesis differentially affects microtubules, and potentially lysosome dynamics.

## Disparate ROS effects on lysosome motility

To dissect these observations further, we considered that microtubule disruption would impair lysosome motility and/or lysosome fusion. To test this model, we quantified lysosome motility and the ability of lysosomes to fuse with phagosomes. First, we conducted live-cell imaging over 3 and 6 min for RAW macrophages (Fig 7A–7C, S1–S6 Movies) and RPE cells (Fig 7D–7F, S7–S13 Movies), respectively, treated with vehicle or ROS agents. Using these videos, we then extracted lysosome speed, track length, and vectorial displacement as indicators of lysosome motility. To our surprise, $H_2O_2$ was the only ROS agent that reduced lysosome speed, track length and vectorial displacement in RAW and in RPE cells, with the strongest effect on the latter cell type (Fig 7). To understand whether microtubule stability was sufficient to impair lysosome coalescence or affect lysosome motility, we performed a control experiment by treating RAW cells with paclitaxel, a microtubule stabilizing agent [49]. However, paclitaxel did not impair lysosome coalescence caused by apilimod and may actually enhance lysosome motility indicators (S5 Fig, S14–S16 Movies), suggesting that $H_2O_2$ blocks apilimod-mediated lysosome enlargement via a distinct mechanism, perhaps by displacing motors from lysosomes or impairing motor activity. If so, this does not seem to occur by reducing the levels of GTP-Rab7 or Arl8b GTPase loaded onto lysosomes as measured by imaging and membrane fractionation (S6 Fig).

Given the impaired lysosome motility caused by $H_2O_2$, we next sought to determine if $H_2O_2$ also hindered lysosome fusogenecity by examining phagosome-lysosome fusion as a model. RAW cells were treated with $H_2O_2$ or vehicle for 1 h, followed by phagocytosis of bacteria for 20 min and a chase period of 40 min to permit phagosome maturation. The degree of phagosome-lysosome fusion was assessed by quantifying the amount of LAMP-1 fluorescent signal present on bacteria-containing phagosomes. We observed that $H_2O_2$-treated RAW cells had less LAMP-1 fluorescence signal localized to bacteria-containing phagosomes compared to vehicle-treated RAW macrophages (Fig 8). This suggests that $H_2O_2$ impaired the ability of lysosomes to fuse with target organelles, consistent with reduced lysosome motility. Overall,

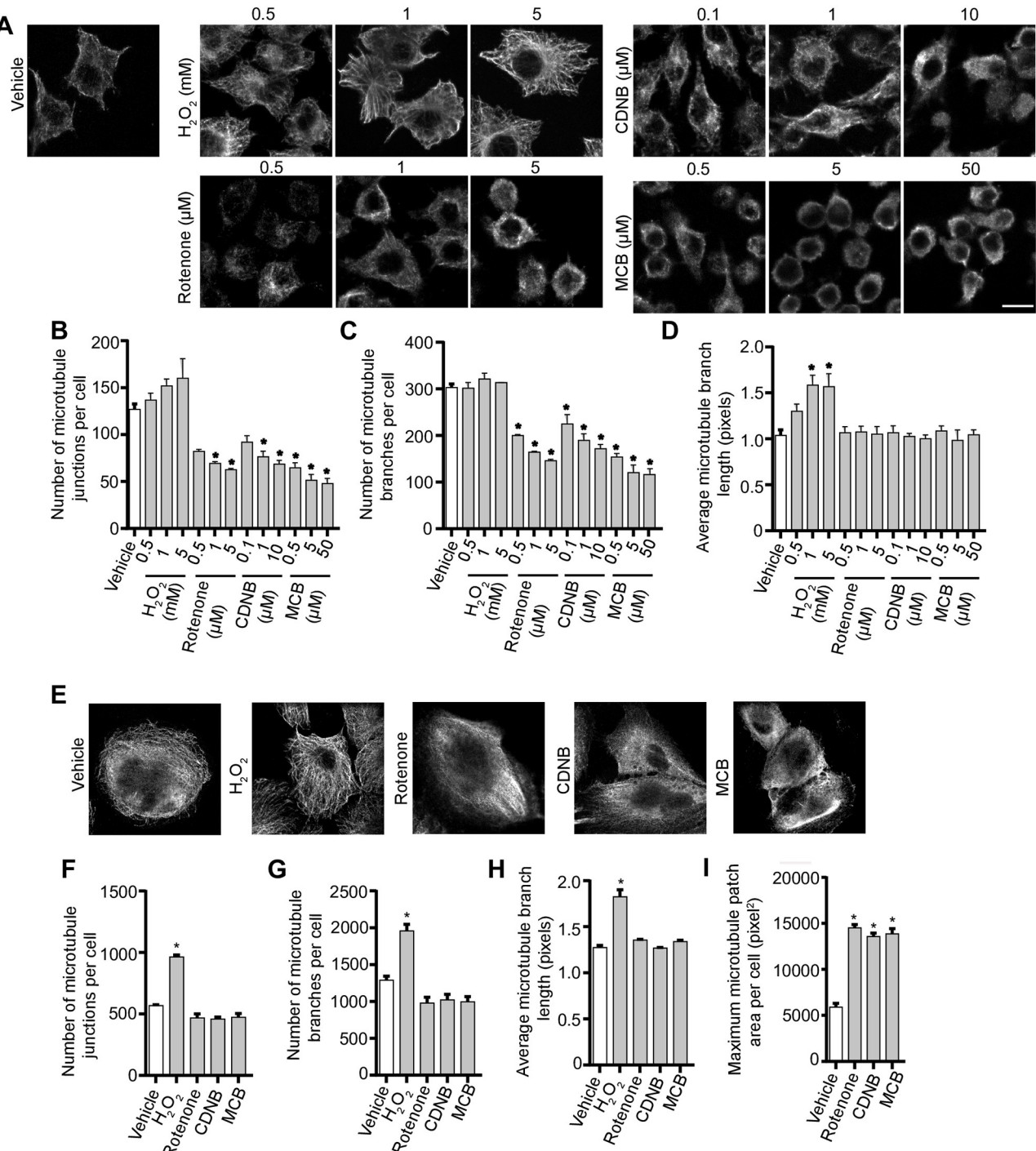

**Fig 6. ROS agents differentially affect the microtubule system.** Representative single z-focal plane immunofluorescence micrographs of RAW cells (A) or RPE cells (E) treated with vehicle, $H_2O_2$, rotenone, CDNB or MCB at previously used time periods and at the indicated concentrations. After treatment with ROS agents, cells were fixed and immunostained with anti-$\alpha$-tubulin antibodies. Quantification of number of microtubule junctions per cell, number of microtubule branches per cell and average microtubule branch length respectively for RAW cells (B-D) and RPE cells (F-H), and patch area in RPE cells (I). Data are represented as mean ± SEM from three independent experiments, with 50–70 cells assessed per treatment per experiment for RAW cells (A-D) and 15–20 cells assessed per treatment per experiment for RPE cells (E-I). One-way ANOVA and Tukey's *post-hoc* test used for B-D and F-I, where * indicates statistically significant difference between control conditions ($P<0.05$). Scale bar: 10 μm (A) or 20 μm (E).

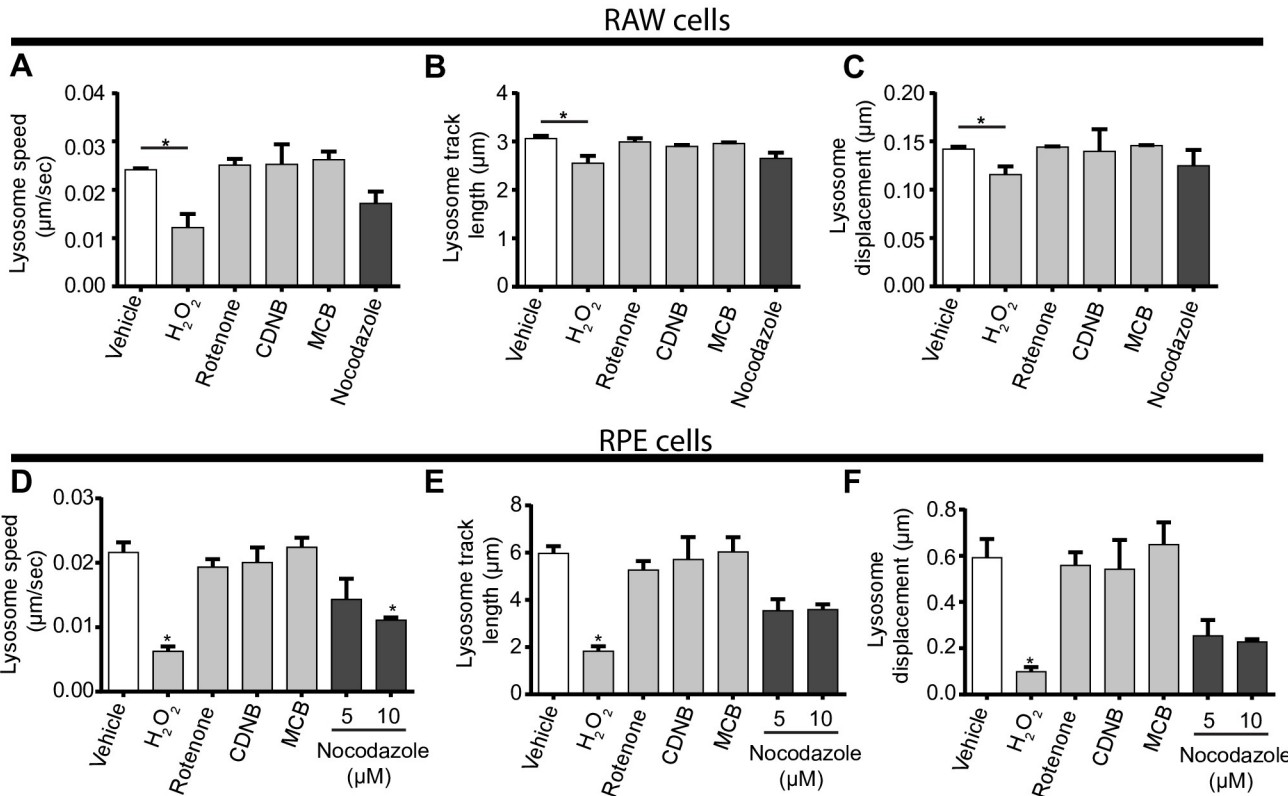

**Fig 7. Distinct ROS agents differentially impact lysosome motility.** RAW cells (A-C, S1–S6 Movies) or RPE cells (D-F, S7–S13 Movies) were pre-labelled with Lucifer yellow and exposed to either vehicle, 1 mM $H_2O_2$ 40 min, 1 μM rotenone 60 min, 10 μM CDNB 30 min, 5 μM MCB 30 min, or 5 μM or 10 μM nocodazole for 60 min. Live-cell spinning disc confocal microscopy was performed at a single, mid-cell z-focal plane once every 4 sec for 3 min for RAW cells or every 8 sec for 6 min for RPE cells. Quantification of lysosome speed (A, D), lysosome track length (B, E), and lysosome displacement (C,F) for RAW cells (A-C) or RPE cells (D-F). Data are represented as mean ± s.d. from three independent experiments. One-way ANOVA and Tukey's *post-hoc* test used for B-D, where * indicates $P<0.05$ between indicated conditions and control. S1–S13 Movies are representative of the live-cell imaging from which shown data was derived from.

we propose that $H_2O_2$ prevents lysosome coalescence during PIKfyve inhibition by impairing lysosome motility and the probability of fusion with other organelles, including phagosomes or other lysosomes.

In comparison to $H_2O_2$, rotenone, MCB and CDNB did not impair measures of lysosome motility in RAW macrophages or RPE cells (Fig 7A–7C) at concentrations sufficient to block apilimod-induced lysosome coalescence. Interestingly, nocadozole strongly impaired all measures in RPE cells but had mild effects on RAW cells (Fig 7). This is likely because RAW macrophages depolymerized for microtubules appeared to become rounder and taller, causing a wobbling motion that moved lysosomes in bulk (see S1–S6 Movies). In comparison, RPE cells were flatter and more resistant to this oscillating effect (see S7–S13 Movies). Given that nocodazole in RPE cells hindered lysosome motility, but CDNB, MCB and rotenone had no effect on lysosome motility measures in RPE cells, this suggests that rotenone, CDNB and MCB only partially disrupt the microtubule system. Thus, the extent of microtubule depolymerization caused by rotenone, CDNB and MCB is not sufficient to explain how these agents prevent lysosome enlargement during apilimod treatment.

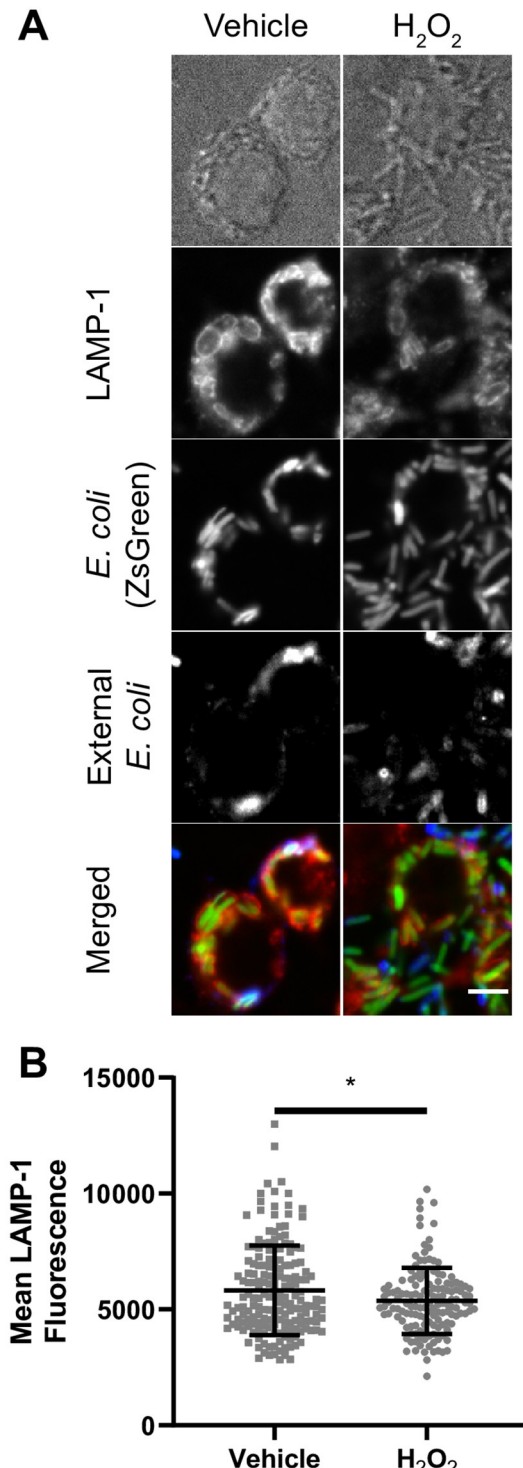

**Fig 8. H₂O₂ hinders phagosome-lysosome fusion.** (A) RAW cells were treated with H₂O₂ or vehicle (H₂O) for 1h before introducing ZsGreen-expressed *E. coli* (green). RAW cells were incubated for 20 minutes in the presence of bacteria and H₂O₂ or vehicle, then RAW cells were washed with PBS, and further incubated in media containing H₂O₂ or vehicle for an additional 40 minutes. External bacteria were labeled with rabbit anti-*E. coli* antibodies (blue) and were excluded from analysis using a mask. LAMP-1 was labeled with rat anti-LAMP-1 antibodies (red). (B) Quantification of mean LAMP-1 intensity on bacteria-containing phagosomes. LAMP-1 intensities were quantified from regions that co-localized to internal bacteria (green signal and no blue signal). Data represented as a scatter plot,

where each dot is an individual phagosome from n = 144 to 179 cells across all independent experiments and conditions. Mean ± standard deviation from three independent experiments is indicated as well. Data analyzed with two-tailed unpaired t-test (* indicates p<0.05).

## ROS effects on lysosome membrane damage, autophagy, degradation, pH and Ca$^{2+}$ release

Given the above, we next assessed if and how ROS affected other lysosome properties in an effort to better understand how some ROS conditions affected lysosome coalescence during PIKfyve inhibition. One possibility is that ROS damage the membrane of lysosomes altering their dynamics. In fact, several works have shown that ROS or drugs that inhibit ROS metabolizing enzymes affect lysosomes. For example, prolonged exposure to the thioredoxin reductase inhibitor, auranofin, accumulates ROS, damages lysosomes, affecting pH and degradation [38, 39]. To test this, we transfected RAW macrophages with galectin-3-GFP, which labels damaged lysosomes with exposed luminal glycoproteins to the cytosol [50, 51]. Strikingly, under the conditions used, $H_2O_2$ or rotenone did not induce a significantly higher number of galectin-3-GFP punctate relative to vehicle-treated cells. As a positive control, we observed higher number of galectin-3-GFP puncta in cells exposed to the lysosome damaging agent, LLMeO (Fig 9A and 9B). Thus, we suspect that membrane damage cannot account for the broad ROS-mediated prevention of lysosome enlargement during PIKfyve inhibition. Second, since ROS have been shown to affect autophagy [33, 52–54], we asked if altered autophagy and/or autophagosome flux during PIKfyve inhibition and ROS exposure might then affect lysosome coalescence. However, using mCherry-GFP-LC3B biosensor to quantify autophagosome formation and flux, we did not observe changes in autophagosome formation or flux by apilimod and/or ROS agents within the 1 h of treatment, time sufficient to alter lysosome size (Fig 9C and 9D). We note that this time frame is typically shorter than conditions used by others when studying PIKfyve, ROS, and autophagy (eg. [33, 55, 56]). Perhaps consistent with the lack of effect on autophagy, apilimod alone did not induce ROS as measured by CellRox Green fluorescence, nor was there any combinatorial effect when cells were co-treated with apilimod and $H_2O_2$ or rotenone (Fig 9E and 9F)–this despite, prior observations that prolonged PIKfyve inhibition elicits ROS in dendritic cells [57].

Third, we then considered if ROS could change lysosomal proteolysis, lysosomal pH, or trigger release of lysosomal $Ca^{2+}$, all of which could in turn could alter lysosome dynamics [25, 58]. In fact, ROS have been connected to release of lysosomal $Ca^{2+}$ via MCOLN1 [54]. Lysosome degradation was tracked by Magic Red cathepsin sensor, while pH was measured by Lysotracker fluorescence. Using acute apilimod inhibition as before, we did not observe a significant change in proteolysis or lysosomal pH relative to vehicle-treated cells (Fig 10). Similarly, adding rotenone or $H_2O_2$-alone did not significantly change these properties. However, co-treatment of cells with apilimod and rotenone or apilimod and $H_2O_2$ did significantly reduce the intensity of Magic Red, suggesting abated proteolytic activity (Fig 10A–10C). However, there was no measurable effect on lysosomal pH (Fig 10D and 10E). Lastly, to examine if lysosomal $Ca^{2+}$ is released by ROS agents, we quantified the lysosome-to-cytosol fluorescence ratio of Fluo4-AM, a $Ca^{2+}$ sensor [59, 60], in rotenone and $H_2O_2$-treated cells. Whereas rotenone had no effect in this ratio, we observed an increase in cytosolic $Ca^{2+}$ levels relative to lysosomes in $H_2O_2$-treated cells (Fig 11A and 11B). However, pre-treating cells with BAPTA-AM did not affect basal lysosome number and volume, or block apilimod-induced lysosome coalescence, nor did it alter $H_2O_2$ or rotenone prevention of lysosome coalescence (Fig 11C–11F).

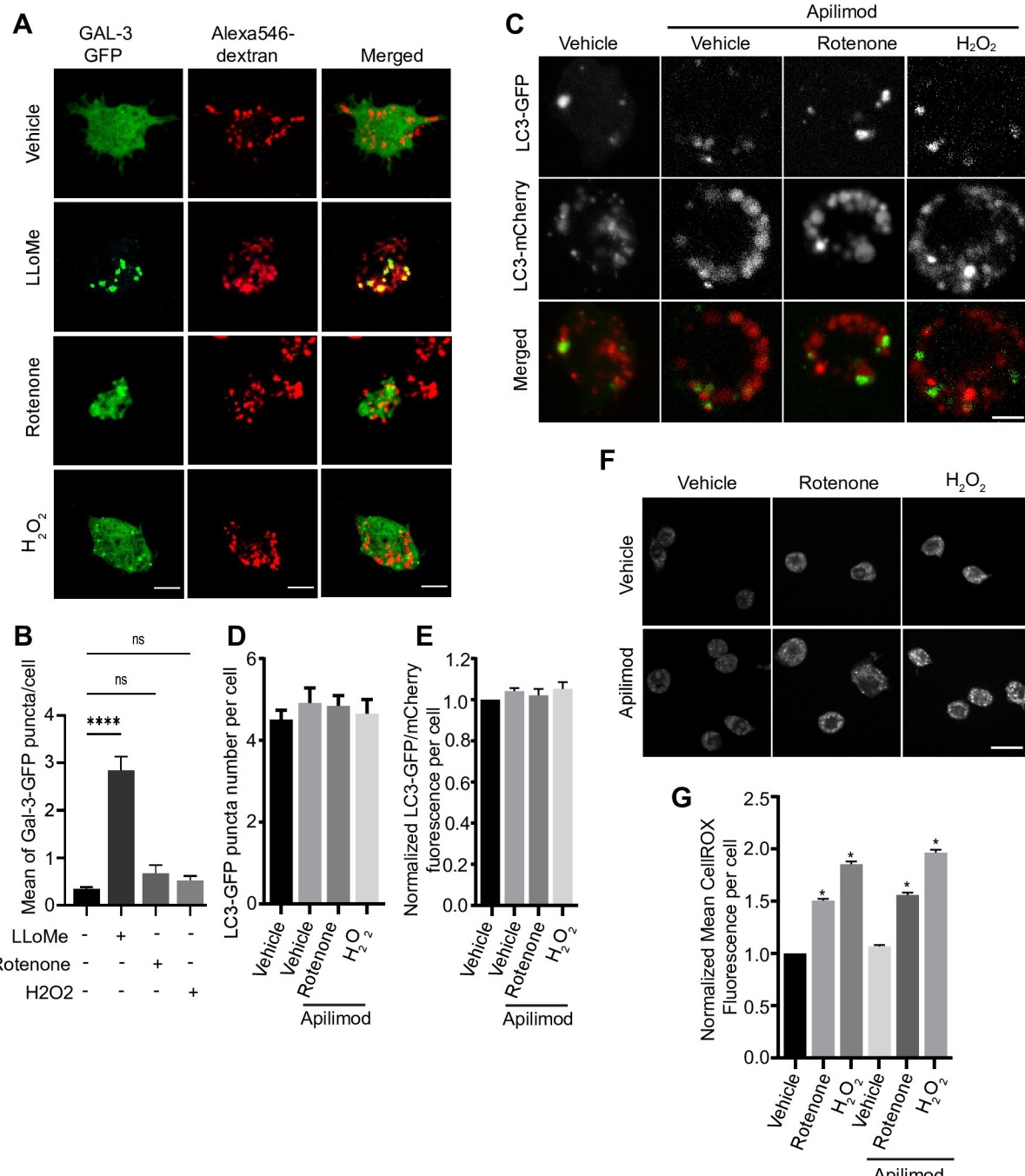

**Fig 9. $H_2O_2$ and rotenone do not induce lysosome damage or alter autophagy.** (A). Macrophages expressing galectin-3-GFP were labelled with Alexa546-conjugated dextran and then exposed to vehicle, rotenone, or $H_2O_2$ for ~ 1 h, or with LLOMe for 2 h as a positive control. Live-cell imaging was done by spinning disc confocal. Scale bar = 10 μm. (B) Mean number of galectin-3-GFP puncta per cell based on 25–30 cells per condition per experiment from n = 3 independent experiment. (C) RAW cells transiently expressing mCherry-eGFP-LC3 and treated with vehicle or with 20 nM apilimod for 40 min alone, with or without 1 μM rotenone for 60 min or 1 mM $H_2O_2$ for 40 min. (D, E) Quantification of the number of LC3-GFP puncta per cell (D) and the ratio of GFP/mCherry intensity (E) based on 25–30 cells per condition per experiment from n = 3 independent experiment. (F) RAW cells treated with vehicle or 20 nM apilimod for 40 min with or without with 1 μM rotenone for 60 min or 1 mM $H_2O_2$ for 40 min, and stained for ROS stimulation with 5 μM CellROX Green for 30 min. (G) Quantification of CellROX Green intensity per cell based on 40–50 cells per condition per experiment from n = 3 independent experiment. One-way ANOVA and Tukey's *post-hoc* test was used, where * indicates $P<0.05$ in relation to vehicle condition and ns indicates not significant.

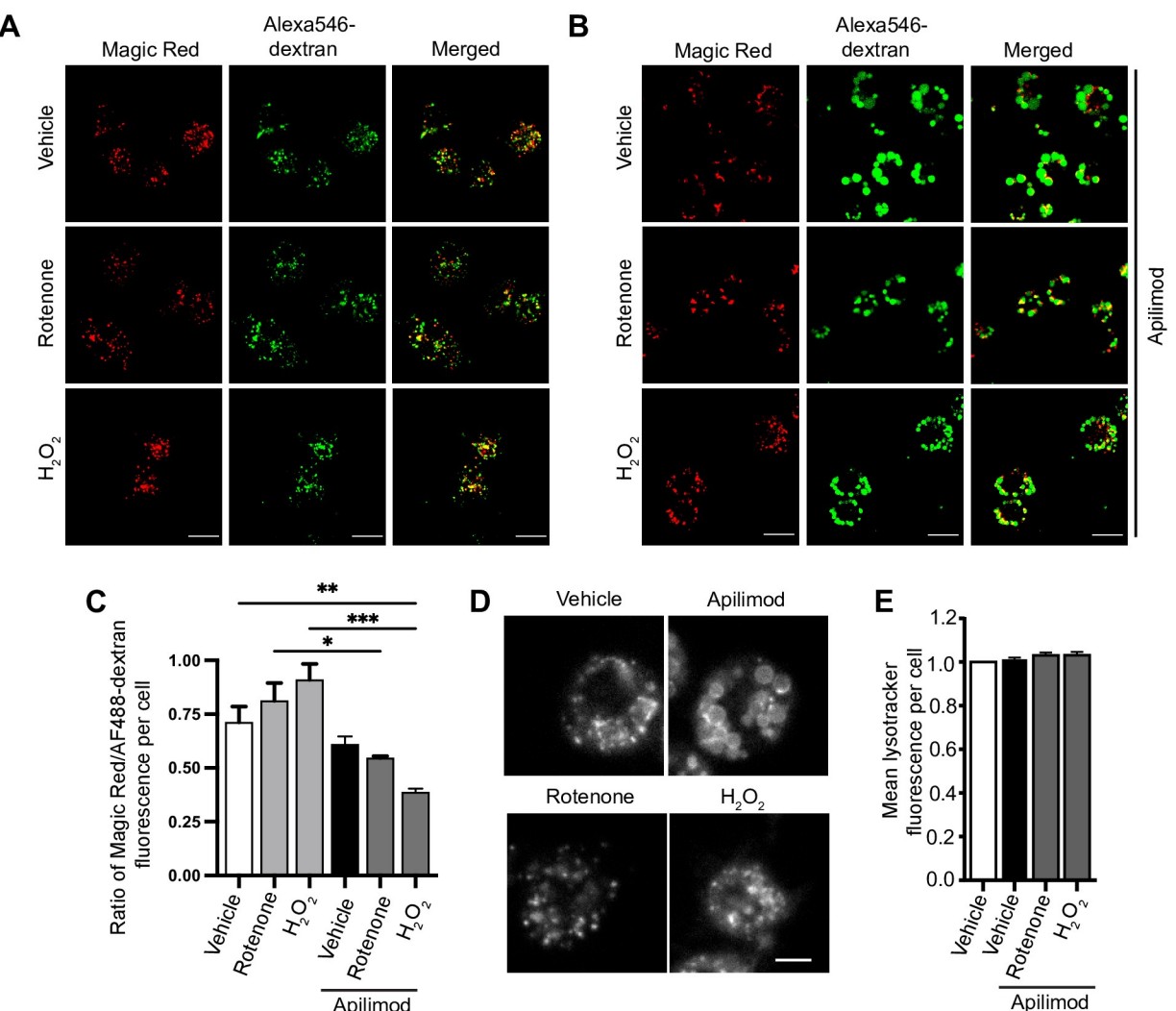

**Fig 10. Measurement of lysosome proteolytic and acidic functions in response to PIKfyve inhibition and ROS.** (A, B) RAW cells loaded with Alexa488-conjugated dextran to label lysosomes followed by treatment with vehicle, or 1 mM $H_2O_2$ for 40 min, or 1 μM rotenone for 60 min (A) or in combination with 20 nM apilimod for 40 min (B), and staining for lysosomal proteolytic function with Magic Red. Scale bar: 50 μm. (C) Quantification of Magic Red to Alexa488-dextran intensity ratio, where decreasing intensity ratio indicates reduced degradation. (D) RAW cells treated with vehicle or 20 nM apilimod for 40 min alone, or in combination with 1 μM rotenone for 60 min or 1 mM $H_2O_2$ for 40 min, and stained for LysoTracker Red as an indicator of lysosome acidification. Scale bar: 5 μm. (E) Quantification of mean lysotracker intensity per cell normalized to control cells. Data are represented as mean ± SEM from three independent experiments, with 40–50 cell assessed per treatment condition per experiment. One-way ANOVA and Tukey's *post-hoc* test was used, where * indicates statistically significant difference between control conditions ($P < 0.05$).

Overall, the data suggest that the effects by ROS on lysosome size are not likely mediated by changes in autophagy, lysosomal damage, $Ca^{2+}$, or pH.

## Clathrin and dynamin are not required for ROS-induced lysosome fragmentation

Lysosomes and related organelles such as autolysosomes can assemble fission machinery, including the canonical fission components, clathrin and dynamin [10, 23, 24, 61]. We sought to determine if ROS species like $H_2O_2$ stimulate clathrin and dynamin-2 to boost fission and prevent lysosome coalescence during PIKfyve inhibition. First, we observed no changes in the

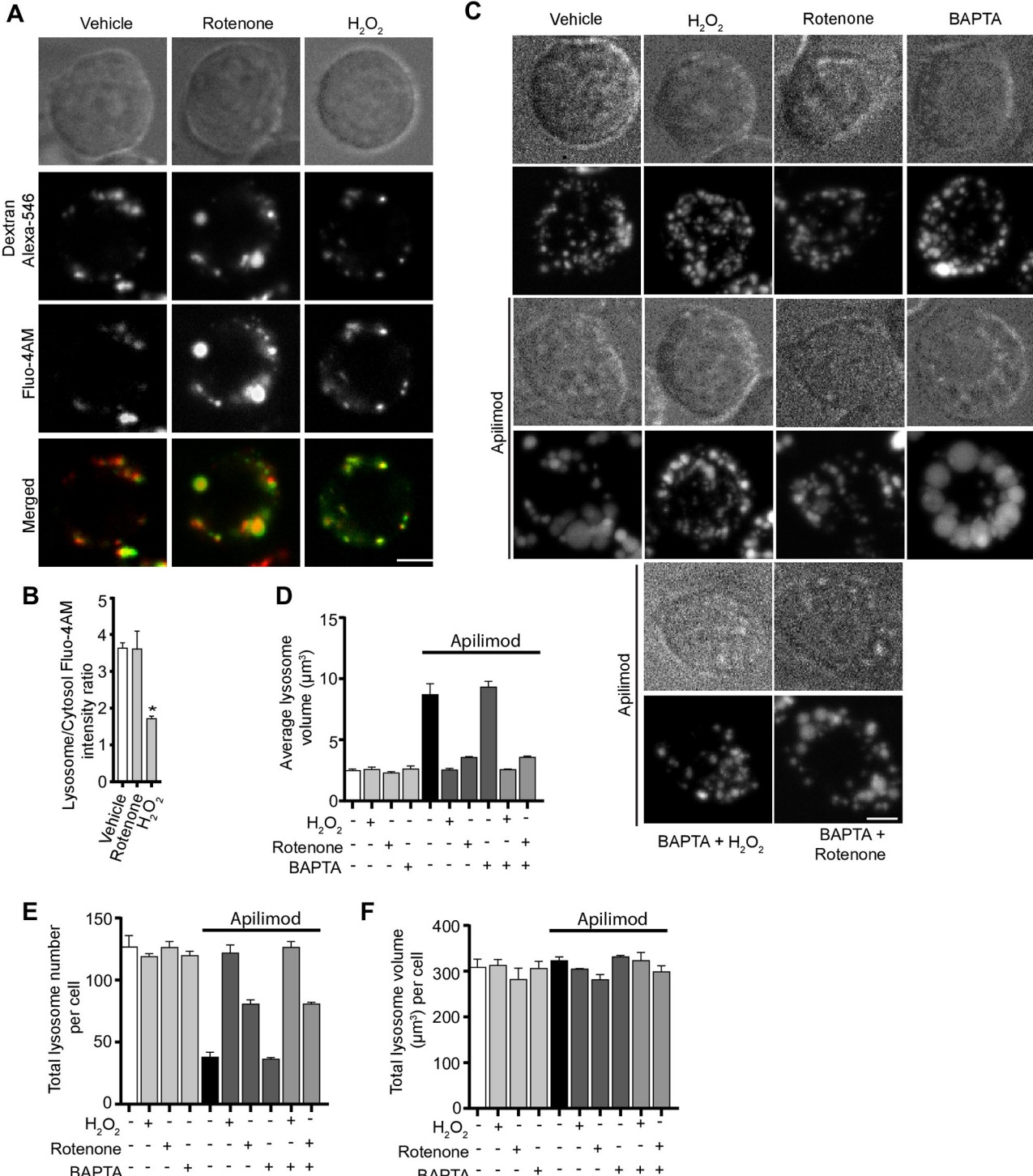

**Fig 11. ROS regulate lysosome coalescence independently of $Ca^{2+}$.** (A) RAW cells were pre-labelled with Alexa546-conjugated dextran, followed by labelling with the $Ca^{2+}$ sensor, Fluo4-AM. Cells were treated with vehicle, or 1 mM $H_2O_2$ for 40 min, or 1 μM rotenone for 60 min. Fluorescence micrographs represent single z-plane images obtained by spinning disc microscopy. Scale bar: 5 μm. (B) Ratio of Fluo4-AM fluorescence intensities associated with Alexa[546]-conjugated dextran and cytosol. Data are represented as mean ± SEM from three independent experiments, with 20–25 cells assessed per treatment per experiment. Two-way ANOVA and Tukey's *post-hoc* test were used for (B), where * indicates $P<0.05$ against control conditions. (C) RAW cells were pre-labelled with Lucifer yellow and exposed to either vehicle, 1 mM $H_2O_2$ for 40 min, 1 μM rotenone for 60 min, or 50 μM BAPTA-AM for 70 min, with or without 20 nM apilimod. Additionally, cells were co-treated with BAPTA-AM and $H_2O_2$ or BAPTA-AM and rotenone, before adding apilimod. Fluorescence micrographs are spinning disc microscopy images with 45–55 z-planes represented as z-projections.

Scale bar: 5 μm. (D-F) Quantification of individual lysosome volume (D), lysosome number per cell (E), and total lysosome volume per cell (F). Data is illustrated as mean ± SEM from three independent experiments, with 25–30 cell assessed per condition per experiment. One-way ANOVA and Tukey's *post-hoc* test used for B-D with *$P<0.05$ compared to indicated control conditions.

levels of lysosome-associated clathrin-eGFP in vehicle or apilimod-treated RPE cells. In comparison, clathrin-eGFP was recruited at higher levels to lysosomes labelled with Alexa[546]-conjugated dextran after $H_2O_2$ treatment in both the presence or absence of apilimod (S7A and S7B Fig). To complement our observations, we treated RAW 264.7 cells with vehicle or $H_2O_2$, followed by sucrose gradient ultracentrifugation to fractionate organelles and probed for clathrin and dynamin by Western blotting. We saw a consistent increase in the level of clathrin and dynamin-2 to LAMP1-positive lysosome fractions in cells treated with $H_2O_2$ relative to resting cells (S7C–S7E Fig). To test whether this enhanced recruitment of clathrin and dynamin aided in lysosome fragmentation during PIKfyve reactivation in the presence of $H_2O_2$, we inhibited clathrin and dynamin with ikarugamycin and dyngo-4a, respectively [62, 63]. Nevertheless, there was no significant difference in the $H_2O_2$-mediated rescue of lysosome coalescence during PIKfyve reactivation when clathrin or dynamin were arrested (Fig 12, S8 Fig). Similarly, there was no difference in lysosome fragmentation during rotenone exposure when cells were incubated with dyngo-4a (S8 Fig). Overall, while at least $H_2O_2$ seems to recruit clathrin and dynamin to lysosomes (and perhaps other membranes), our data do not support a role for clathrin and dynamin in preventing lysosome coalescence during PIKfyve inhibition under the used conditions.

## ROS prevents lysosome coalescence by actin depolymerization

There is growing evidence that F-actin-based structures may regulate endosomal and lysosomal fission, either through the action of acto-myosin constriction or the assembly of fission machinery that remains to be fully defined [10, 21, 64]. In fact, work by Hong et al. suggests that PIKfyve inhibition causes branched actin accumulation on endosomes; based on their markers used to identify endosomes, lysosomes were likely included in their analysis [21]. We set to understand if at least some ROS can prevent lysosome coalescence during PIKfyve inhibition by eliminating these F-actin assemblies on lysosomes. Indeed, PtdIns(3,5)P$_2$ depletion increased the number of F-actin puncta associated with lysosomes detectable by fluorescent-phalloidin staining (Fig 13A and 13B), as previously reported [65, 66]. Interestingly, co-administration of rotenone or CDNB with apilimod reduced F-actin puncta associated with lysosomes (Fig 13A and 13B). These observations indicate that ROS generated by rotenone and CDNB help prevent or reverse lysosome coalescence during PIKfyve inhibition by boosting actin turnover on lysosomes.

To further test whether actin depolymerization helps prevent lysosome coalescence during PIKfyve inhibition and accelerate lysosome fragmentation during PIKfyve inhibition, we compared lysosome volumetrics in cells treated with the actin depolymerizing agents, cytochalasin B or latrunculin A. We found that both cytochalasin B and latrunculin A treatments hindered lysosome coalescence during apilimod treatment, as well as accelerated lysosome fragmentation after apilimod removal and PIKfyve reactivation (Fig 13C–13F). Collectively, our observations suggest that at least certain types of ROS prevent lysosome coalescence during acute PIKfyve inhibition by alleviating F-actin amassed on lysosomes, likely facilitating fission.

## Discussion

Low PtdIns(3,5)P$_2$ levels causes multiple defects including impaired autophagic flux, nutrient recycling, and phagosome resolution [10, 27]. These defects are likely derived from the

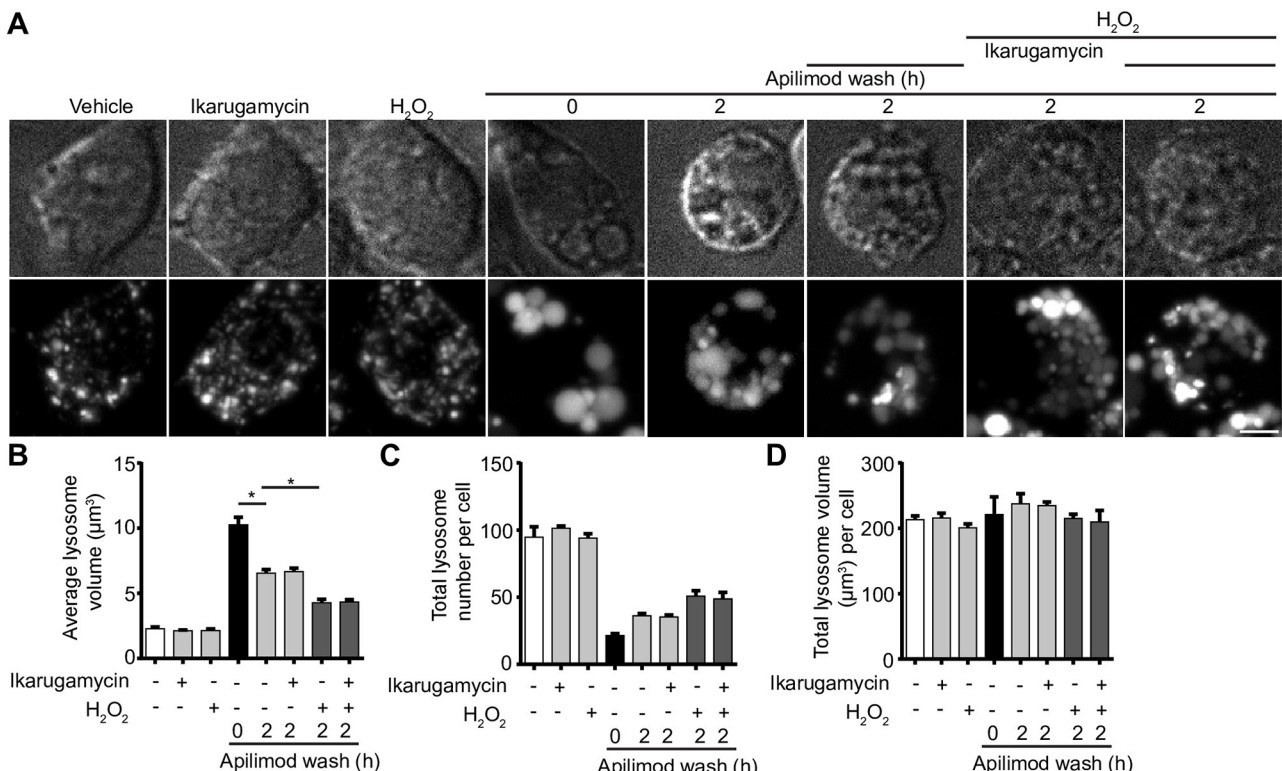

**Fig 12. Clathrin inhibition does not arrest ROS-mediated lysosome fragmentation during PIKfyve reactivation.** (A) RAW cells were pre-labelled with Lucifer yellow and exposed to either vehicle alone, 1 mM $H_2O_2$ for 40 min, 1 μM ikarugamcyin for 1 h, or 20 nM apilimod for 60 min. For a subgroup of cells treated with apilimod, drug was replaced with fresh media containing either vehicle, 1 mM $H_2O_2$, 1 μM ikarugamcyin, or 1 mM $H_2O_2$ and 1 μM ikarugamcyin for 2 h. Fluorescence micrographs are spinning disc microscopy images with 45–55 z-planes represented as z-projections. Scale bar: 5 μm. (B-D) Quantification of individual lysosome volume (B), lysosome number per cell (C), and total lysosome volume per cell (D). Data are shown as mean ± s.e.m. from three independent experiments, with 25–30 cell assessed per treatment condition per experiment. One-way ANOVA and Tukey's *post-hoc* test used for B-D, where * indicates $P<0.05$ between experimental and control conditions.

inability of lysosomes to reform or separate after fusion with other lysosomes, late endosomes, phagosomes, and autolysosomes [4, 10, 30, 67, 68]. As a corollary, lysosomes coalesce to become larger but fewer [4, 30]. Thus, identification of mechanisms or compounds that can drive lysosome fission may prove useful to rescue autophagic flux, degradative capacity, and lysosome dynamics in cells exhibiting reduced PtdIns(3,5)P$_2$ levels. Such mechanisms or compounds may act to up-regulate PtdIns(3,5)P$_2$ levels in conditions of insufficient PIKfyve activity like those caused by null-mutations in the Fig 4 lipid phosphatase [29]. For example, the cyclin/cyclin-dependent kinase, Pho80/Pho85, phosphorylates Fab1 to upregulate the levels of PtdIns(3,5)P$_2$ in response to hypertonic shock, protecting yeast cells from osmotic shock [69, 70]. Alternatively, activating mechanisms downstream of PtdIns(3,5)P$_2$ that enable lysosome fission directly may also rescue lysosome dynamics.

We previously observed that photo-toxicity during live-cell imaging with spinning disc confocal microscopy prevented lysosome coalescence during apilimod-mediated PIKfyve inhibition [30]. While unfortunately blunting our ability to perform high spatio-temporal resolution of lysosome enlargement by live-cell imaging, we questioned if other sources of ROS could also prevent lysosome coalescence during PIKfyve inhibition. Indeed, we provide evidence here that ROS generated by diverse approaches can counteract and help reverse lysosome coalescence during PIKfyve inhibition. Notably, neither $H_2O_2$ or rotenone rescued

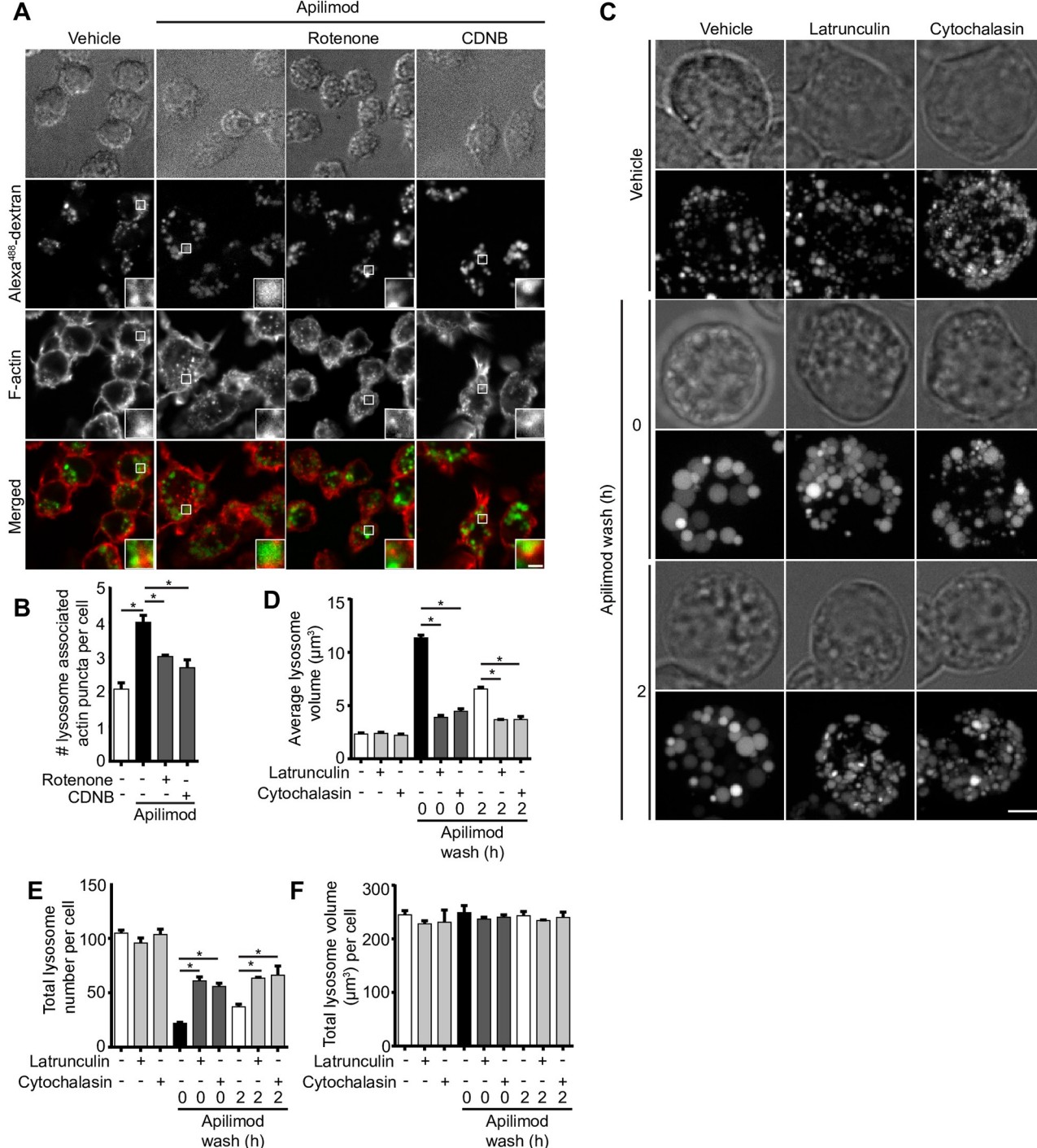

**Fig 13. ROS promote actin clearance from lysosomes and actin depolymerization abates lysosomes coalescence during PIKfyve inhibition.** (A) RAW cells pre-labelled with Alexa[488]-conjugated dextran followed by treatment with vehicle, 20 nM apilimod for 40 min alone, or in presence of 10 μM CDNB for 30 min or 1 μM rotenone for 60 min. Cells were fixed with 4% PFA and stained for actin with phalloidin. Fluorescence micrographs were captured by spinning disc confocal as single z-planes. The inset is a magnified portion of field of view tracking Alexa[488]-conjugated dextran lysosome (s), phalloidin-stained actin, and as merged channels. Scale bar: 2 μm. (B) Cells were assessed for number of actin puncta structures associated with lysosomes. Data represent mean ± S.E.M. from three independent experiments, with 60–80 cells assessed per treatment condition across three experiments. One-way ANOVA and Tukey's *post-hoc* test was used, where * indicates statistical significance between indicated conditions ($p<0.05$). (C) RAW cells were pre-labelled with Lucifer yellow and exposed to vehicle or 20 nM apilimod for 1 h, followed by apilimod removal at 0 or 2 h. These conditions were then supplemented with additional vehicle or 1 μM latrunculin A or 5 μM cytochalasin B for 1 h. Fluorescence micrographs are represented as z-projections of 45–55 z-plane images obtained by spinning disc confocal microscopy. Scale bar: 5 μm. D-F: Quantification of individual

lysosome volume (D), lysosome number per cell (E), and total lysosome volume per cell (F). Data represent mean ± S.E.M. from three independent experiments, with 25–30 cells assessed per treatment condition per experiment. One-way ANOVA and Tukey's *post-hoc* test was used, where * indicates statistical significance between indicated conditions ($p<0.05$).

lysosome size by up-regulating the levels of PtdIns(3,5)$P_2$. This suggests that ROS counteract lysosome coalescence by acting downstream of PtdIns(3,5)$P_2$, or by stimulating parallel processes that promote lysosome fission or impair lysosome fusion. Notably, ROS agents alone did not appreciably alter basal lysosome properties like lysosome size and number. This may partially relate to resolution limit of light microscopy as we estimate the radius of intact lysosomes to be 0.7 μm, or due to physical constraints of lysosomes that prevent smaller average lysosome size, or perhaps because the mechanisms responsible for basal lysosome dynamics are insensitive to ROS effects. Collectively, our work suggests that ROS prevent lysosome enlargement during PIKfyve inhibition, but exact mode of action may depend on ROS type, and/or mode of production, and/or location since $H_2O_2$ had distinct effects from those by CDNB, auranofin, MCB and rotenone. In fact, under certain conditions, PIKfyve inhibition has been shown to accumulates ROS, while lysosome enlargement proceeds [57].

We previously showed that disrupting the microtubule system with nocodazole or impairing motor proteins abated lysosome coalescence during PIKfyve inhibition [30]. Thus, we explored whether ROS agents disrupted lysosome motility, which would impair fusogenicity. We observed that only $H_2O_2$ significantly arrested lysosome motility and reduced fusogenicity. In addition, we also saw a more defined microtubule network in $H_2O_2$-treated cells, suggesting that microtubules were stabilized by $H_2O_2$. Whether this effect is generalizable is debatable since there are contradictory observations about the effect of $H_2O_2$ on the microtubule system, which may depend on cell type and experimental conditions employed [71–73]. Regardless, microtubule stabilization is not sufficient to impair lysosome motility since paclitaxel did not prevent lysosome enlargement caused by apilimod or impair lysosome motility. Thus, we propose that $H_2O_2$ impairs lysosome motility by disrupting motor activity, and/or impairing motor interactions with lysosomes and/or microtubules. Mitochondria may offer some insight since their motility was also arrested by $H_2O_2$ [72]. $H_2O_2$ stimulated p38α MAPK, which then interrupted motor adaptor complex function. Motors themselves retained their activity since forced anchorage of kinesin to mitochondria maintained mitochondrial motility in the presence of $H_2O_2$ [72]. Thus, $H_2O_2$ may disrupt kinesin and/or dynein adaptors like RIL-P-ORP1L-dynein or SKIP-kinesin, though our work suggests that this is not likely occurring by altering GTP-Rab7 and Arl8b loading onto lysosomes.

In comparison to $H_2O_2$, ROS produced by mitochondrial uncoupling (rotenone) or amassed by disrupting catalase (MCB), thiol group inhibition (CDNB), or thioredoxin reductase (auranofin) partially depolymerized the microtubule system under the employed conditions, though not sufficiently enough to hinder lysosome motility. Instead, these agents may prevent overt lysosome coalescence by releasing a dense F-actin network that assembles on lysosomes during PIKfyve inhibition. Consistent with this, actin depolymerizers also reduced lysosome coalescence during acute PIKfyve inhibition and promoted lysosome fragmentation during PIKfyve reactivation. These observations may be consistent with those by Hong *et al.*, wherein PtdIns(3,5)$P_2$ modulates branched actin dynamics on endosomes (using markers that can apply to lysosomes as well) by regulating cortactin [21]. The authors showed that PIKfyve inhibition increased actin density on endo/lysosomes, which consequently impaired fission and caused enlargement [21]. Additionally, PIKfyve was recently shown to modulate branched F-actin to help drive melanosome maturation [65]. Indeed, branched F-actin has emerged as a major player in membrane fission for endo/lysosomes nucleated by ER-endosomes contact

sites [74, 75]. Thus, we propose that ROS generated by rotenone, CDNB, auranofin, and perhaps MCB, may abate lysosome coalescence by relieving dense F-actin networks that form on lysosomes during PIKfyve inhibition.

While oxidative stress in cells can hinder lysosome coalescence during PIKfyve inhibition and promote lysosome fragmentation during PIKfyve reactivation, the exact mechanisms of action depends on the type of ROS and/or mode of production. Of the treatments we employed, $H_2O_2$ was able to produce cytosolic and mitochondrial superoxide $O_2^-$ and $OH^.$ radicals, but no detectable levels of singlet $O_2^.$ (Fig 1); likely, the ROS generated was delocalized as well. In comparison, rotenone and CDNB released singlet $O_2^.$, while rotenone also released mitochondrial superoxide and singlet $O_2^-$; neither treatment appeared to produce $OH^.$ radicals (Fig 1). We propose that the likely delocalized release of ROS and $OH^.$ generated by $H_2O_2$ may stabilize microtubules and impair motor dynamics, which then impinges on lysosome dynamics [76–78]. In comparison, mitochondrial $O_2^-$ and singlet $O_2^.$ increases turnover of the lysosomal F-actin network to shift lysosome dynamics towards fission. While we could not detect specific ROS in MCB, MCB interacts with GSH and with thiol residues of other enzymes such as thioredoxin reductase, leading to increased $O_2^-$ production [43, 77, 79]. It also remains possible that not all ROS species impact lysosome fusion-fission cycle. In addition, ROS may affect lysosome fission-fusion cycles differentially depending on where they are being generated and their level. For example, it is conceivable that exogenous ROS like $H_20_2$ or formed by mitochondria may impact lysosome dynamics, whereas ROS generated by PIKfyve inhibition itself as observed in dendritic cells may impact cells differently [57]. Overall, future work should aim to better delineate the impact that the type of ROS and location of generation have on lysosome dynamics, and ultimately to identify their molecular target responsible for altering lysosome dynamics. Since ROS can serve as physiological signals [37, 80], it is tempting to suggest that particular ROS may play a role in coordinating localized processes like membrane fusion, motor activity, and membrane fission. This process or the sensors engaged by ROS may represent approaches to rescue lysosome dynamics in conditions of PtdIns(3,5)P$_2$ insufficiency.

## Materials and methods

### Cell culture, plasmids, and transfection

RAW 264.7 macrophages and HeLa cells were grown in Dulbecco's Modified Eagle Medium (DMEM; Wisent, St Bruno, QC) supplemented with 5% heat-inactivated fetal bovine serum (FBS; Wisent). ARPE-1 (RPE) cells stably expressing clathrin light chain-eGFP were grown in DMEM/F12 medium (Gibco) supplemented with 10% FBS [81]. All cells were grown at 5% $CO_2$ and 37˚C and routinely checked for contamination. FuGene HD (Promega, Madison, WI) was used for transient transfections following manufacturer's instructions with a ratio of 3:1 FuGene HD transfection reagent (μl) to DNA (μg). The transfection mixture was replaced with fresh complete medium 4–5 h post-transfection and cells were used 24 h following transfection. RAW cells were transfected with plasmids expressing Rab7-RILPC33-GFP (RILPC33-GFP), or wild-type Arl8b-GFP (Arl8bWT-GFP), previously described in [13, 82], or Galectin-3-GFP (Addgene; [50]), or mCherry-eGFP-LC3B (plasmid #22418; [83]). The bacterial expression vector pZsGreen (Takara Bio USA, Inc., formerly Clontech Laboratories, Inc., 632446) was transformed into E. coli DH5α to generate ZsGreen-expressing bacteria.

### Pharmacological treatment of cells

Apilimod (Toronto Research Chemicals, Toronto, ON) was used at 20 nM for 40 min, unless otherwise indicated, to deplete cellular PtdIns(3,5)P$_2$. $H_2O_2$ (Bio Basic, Markham, ON) was

used as indicated. Rotenone, 1-chloro-2,4,-dinitrobenzene (CDNB), auranofin, and monocho-lorobimane (MCB; all from Sigma-Aldrich, Oakville, ON) were used as indicated to generate ROS by respectively inhibiting mitochondrial respiratory chain complex, thiol group, thiore-doxin reductase, or glutathione. Bovine liver catalase (Sigma-Aldrich) and N-acetyl-L-cysteine (NAC) (Bio Basic) were used as anti-oxidants. Paclitaxel and nocadozole (both from Sigma-Aldrich) were used at 1 or 10 μM and 5 or 10 μM to stabilize and depolymerize microtubules, respectively. Latrunculin A (Abcam, Toronto, ON) and cytochalasin D (EMD Millipore, Toronto, ON) were used at 1 μM and 5 μM, respectively to depolymerize actin. Ikarugamycin (Sigma-Aldrich) and dyngo-4A (Abcam, Cambridge, MA) used to inhibit clathrin and dyna-min respectively. BAPTA-AM (Sigma-Aldrich) was used to chelate intracellular calcium and Fluo4-AM (ThermoFisher, Burlington, ON) was used as a fluorescent $Ca^{2+}$ probe. As a posi-tive control for lysosome damage, we treated cells for 2 h with 0.5 mM L-leucyl-L-leucine methyl ester (LLOMe; L7393, Sigma-Aldrich).

## Lysosome labelling

Lysosomes were labelled by incubating cells with 200 μg/mL Alexa[546]-conjugated dextran or with 200 μg/mL Alexa[488]-conjugated dextran (Thermo Fisher Scientific, Mississauga, ON) or with 2.5 mg/mL Lucifer yellow (Thermo Fisher Scientific, Mississauga, ON) for 2 h in com-plete media at 37˚C in 5% $CO_2$. Cells washed with phosphate-buffered saline (PBS) and resup-plied with complete cell-specific media for 1 h to chase the fluid-phase marker to lysosomes before pharmacological manipulation and live-cell imaging. We note that we use "lysosomes" to represent a potential mixture of late endosomes, lysosomes and endolysosomes [5, 30]. Lysosomal calcium was labelled with Fluo-4AM 8 μM by pulsing for 45 min in complete media at 37˚C in 5% $CO_2$, followed by washing with PBS and addition of complete media for 45 min to chase the marker to lysosomes.

## Live- and fixed-cell spinning disc confocal microscopy

Microscopy and imaging were done with a Quorum DisKovery spinning disc confocal micro-scope system equipped with a Leica DMi8 microscope connected to an iXON 897 EMCCD camera, controlled by Quorum Wave FX powered by MetaMorph software, using 63x 1.4 NA oil-immersion objective (Quorum Technologies, Guelph, ON). Live-cell imaging was per-formed using environmental chamber set to 5% $CO_2$ and 37˚C in complete cell-specific medium. Standard excitation and emission filter sets and lasers were used for all fluorophores. RAW and HeLa cells, unless otherwise indicated, were imaged as z-projections of 45–55 z-planes with 0.3 μm distance between each plane, or 20–30 z-planes with 0.3 μm distance between each plane for RPE cells, as acquired by spinning disc confocal microscopy. For time-lapse imaging, RAW cells were imaged using single, mid-section z-plane every 4 s for 3 min. RPE cells were imaged using single, mid-section z-plane every 8 s for 6 min. Clathrin-eGFP expressing RPE cells were imaged every 2 min for 40 min.

## Detection of ROS production

For determining intracellular net ROS production, we incubated RAW 264.7 macrophages with 5 μM of the cell-permeable redox sensitive dye, CellROX Green (Thermo Fisher Scien-tific), for 30 min at 37˚C with 5% $CO_2$ in the dark during treatment with various ROS pro-ducing agents. Cells were washed twice with PBS followed by replenishment with complete media and imaging. Light-induced ROS production was detected by incubating cells with 1 mg/mL of nitroblue tetrazolium (NBT; Thermo Fisher Scientific) for 30 min at 37˚C with 5% $CO_2$ in the dark. To detect specific intracellular ROS, we used several probes:

hydroxylphenyl fluorescein to detect hydroxyl radical and peroxynitrite (HPF; Thermo-Fisher Scientific), MitoSox Red for mitochondrial superoxide (ThermoFisher Scientific), Biotracker Si-DMA for singlet oxygen (Millipore Sigma), and ROS-ID detection kit (Enzo Life Sciences) for general superoxide. After treatment with ROS inducers, cells were washed with PBS 3x before adding these fluorescent probes. Cells were incubated with 5 μM Mito-Sox Red for 10 min at 37 ˚C with 5% $CO_2$ in the dark, or 100 nM Si-DMA, or 10 μM HPF for 45 min. For ROS-ID, 0.06 nM ROS-ID was added 1 h before, incubated at 37 ˚C with 5% $CO_2$ in the dark, followed by washing with PBS and adding ROS inducers. After treatment with ROS probes or inducers, cells were washed 3x PBS and supplemented with probe specific media. All experiments were imaged using live-cell spinning disc confocal microscopy as described.

## Cell viability analysis with Propidium Iodide

RAW cells seeded in DMEM complete media were exposed to ROS agents as indicated and then stained with propidium iodide (ThermoFisher Scientific) according to manufacturer's instructions. Cells were fixed with 4% (v/v) paraformaldehyde in PBS and imaged with spinning disc confocal microscope.

## Immunofluorescence and F-actin imaging

Following experimentation, cells were fixed for 15 min with 4% (v/v) paraformaldehyde in PBS, permeabilized for 10 min with 0.1% Triton X-100 (v/v) in PBS, and then blocked with 3% BSA (v/v) in PBS. Subsequently, cells were incubated with mouse monoclonal antibody against α-tubulin (1:200; Sigma-Aldrich), followed by incubation with donkey Dylight-conjugated polyclonal antibody against mouse IgG (1:1000; Bethyl), and samples were then mounted in Dako mounting media for subsequent imaging. Alternatively, lysosomes were labelled with Alexa[488]-conjugated dextran as before, followed by fixation for 15 min with 4% (v/v) paraformaldehyde, permeabilized for 10 min with 10 μg/ml digitonin (Promega, Madison, WI), and blocked with 3% BSA (v/v), all solutions in PBS. Cells were then stained for F-actin with fluorescent-phalloidin (ThermoFisher Scientific).

## Lysosome damage detected by galectin-3-GFP

RAW cells were seeded in DMEM supplemented with 5% FBS for 24 h at 37˚C in 5% $CO_2$. Cells were transfected with Galectin-3-GFP plasmid (0.5 μg) using FuGene HD (Promega, Madison, WI) with a ratio of 3:1 for 24 h at 37˚C in 5% $CO_2$. Post-transfection, cells were treated with 1 mM $H_2O_2$ for 40 min, 1 μM rotenone for 60 min or for 2 h with 0.5 mM L-leucyl-L-leucine methyl ester (LLOMe; L7393, Sigma-Aldrich).

## Image analysis

To determine lysosome number, individual lysosome volume and total cellular lysosome volume, we used Volocity (Volocity 6.3.0) particle detection and volumetric tools. Z-stack images were imported into Volocity and a signal threshold was applied at 2x the average cytosolic fluorescence intensity. Particles were defined as being greater than 0.3 μm$^3$ for inclusion into the analysis, and if necessary, a watershed function was applied to split lysosome aggregates caused by thresholding. Regions of interest were drawn surrounding individual cells for cell-to-cell analysis. Lysosome speed, track length, and displacement were assessed using Imaris (BitPlane, Concord, MA) with 'ImarisTrackLineage' module.

To determine the level of membrane-bound RILP-C33 and Arl8b, we estimated the membrane-bound to cytosolic ratio of fluorescently-tagged proteins. Using ImageJ, lines that were 3-pixel wide by 20-40-pixel long were assigned to areas of transfected cells using a predetermined grid to avoid bias but excluding the nucleus. Plot profiles were then obtained, exported into an Excel spreadsheet, values were arranged according to fluorescence intensity, and the ratio calculated for highest 10 pixels over lowest 10 pixels along the length of the line ($F_H/F_L$ fluorescence ratio); the expectation is that values approximate to 1 represent low membrane signal due to mostly cytosolic signal, while ratio values greater than 1 represent signal that localizes to punctate structures relative to cytosol (Chintaluri et al., 2018).

For determination of clathrin-GFP on lysosomes, RPE cells stably expressing clathrin heavy chain-eGFP were loaded with Alexa[546]-conjugated dextran and treated with apilimod, followed by imaging with spinning disc confocal microscope. Image analysis was performed using ImageJ by thresholding Alexa[546]-conjugated dextran signal and generating a mask, which was then applied to the green (clathrin) channel to determine the GFP fluorescence intensity on regions marked by dextran signal. Regions of interest within the cytosol and the extracellular space were drawn to respectively obtain mean cytosolic fluorescence intensity and background. These values were then used to calculate the ratio of lysosome-to-cytosol clathrin-eGFP. Similar approach was employed to determine Fluo-4AM intensity for dextran Alexa[546] lysosomal structures over cytosolic Fluo-4AM to obtain lysosome-to-cytosol Fluo-4AM intensity ratio, or to obtain LC3B-GFP/mCherry intensity ratio. To determine the fluorescence of intracellular CellROX Green or other ROS probes or propidium iodide, images were imported onto Volocity (Volocity 6.3.0) or ImageJ, regions of interest were drawn around cell, and mean fluorescence intensity per cell was recorded and background-corrected. Similar approach was employed to determine Fluo-4AM intensity for dextran Alexa[546] lysosomal structures over cytosolic Fluo-4AM to obtain lysosome-to-cytosol Fluo-4AM intensity ratio. To determine the fluorescence of intracellular CellROX Green or other ROS probes, images were imported onto Volocity (Volocity 6.3.0) or ImageJ, regions of interest were drawn around cell, and mean fluorescence intensity per cell was recorded and background-corrected. For galectin-3-GFP analysis, images were imported into ImageJ, background-corrected, and then thresholding was applied to each individual transfected cell (25–30 cells per condition) to identify galectin-3-GFP puncta. Particles ranging between 50–1000 $\mu m^2$ were then counted, with the assumption that smaller size particles corresponded to noise. Similar approach was used to threshold and identify Lysotracker positive particles within a cell, and to measure mean Lysotracker particle intensity per cell.

To assess microtubule structure, we sought to use several measures as proxies for microtubule alteration under different treatments. Single-plane images were converted to 8-bit images through ImageJ followed by application of fluorescence intensity threshold to select microtubules. Images were converted to binary and filaments analyzed through "skeleton" and "Analyzeskeleton". Total number of microtubules junctions, where junctions represent filamentous pixels from where two or more microtubule branches arise, total number of microtubule branches and average microtubule branch length were scored and collected for data analysis. Alternatively, RPE cell microtubule structure was analyzed through applying binary filter to fluorescent microtubules, followed by watershed segmentation to segregate microtubules into areas of tubulin patches with the expectation that depolymerized microtubules pool into large patches compared to intact tubulin.

Image contrast enhancement was performed with Adobe Photoshop CS (Adobe Systems, San Jose, CA) or ImageJ without changing relative signals and applied after quantification. Adobe Illustrator CS (Adobe Systems) was used for constructing figures.

## Lysosome fractionation

RAW 264.7 cells were grown and used according to manufacturer's instructions to obtain membrane fractions by differential sedimentation ultracentrifugation using a density gradient (Lysosome Isolation Kit, Sigma-Aldrich, LYSISO1). Briefly, cells were lysed and homogenates centrifuged 1,000 x*g* for 10 min at 4˚C to separate unbroken cells and debris from cytoplasmic membranes. The supernatant was further centrifuged at 20,000 x*g* for 20 min at 4˚C to pellet lysosomes and other organelles. The pellet was reconstituted with Optiprep density gradient medium (60% (w/v) solution of iodixanol in water and sucrose) and loaded onto of a step-wise sucrose gradient as described by the manufacturer and subjected to ultracentrifugation at 150,000 x*g* for 4 h at 4˚C using SW50.1 rotor (Beckman Coulter, Mississauga, ON). Fractions were then collected and subject to denaturation with Laemmli buffer until further use.

## Membrane fractionation

RAW cells were lysed in 200 µl ice cold homogenization buffer (3 mM imidazole, 250 mM sucrose, 0.5 mM EDTA, pH 7.4 with protease inhibitor cocktail). Cells were homogenized by passing 10x though a 25-gauge needle, then lysates were sequentially centrifuged at 3000 x*g* for 10 min at 4˚C and 7,000 x*g* for 10 min at 4˚C to clear supernatants. Supernatants were then further centrifuged at 100,000 x*g* using SORVALL wX+ULTRA-centrifuge (Thermo Scientific) for 30 min at 4˚C to separate cytosol and membranes. Next, the pellets were resuspended in 0.5% digitonin in solubilization buffer (50 mM NaCl, 50 mM imidazole, 2.5 mM 6-aminohexanoic acid, 2 mM EDTA, pH ~7) to obtain membrane-bound materials.

## Western blotting

For whole-cell lysates in 2x Laemmli buffer, cells were passed six times through 27-gauge needle, heated. Cell lysates or cell fractions were resolved through SDS-PAGE with 10% acrylamide resolving gel. Proteins were then transferred to a PVDF membrane, blocked and incubated with primary and HRP-conjugated secondary antibodies in Tris-buffered saline containing 5% skimmed milk and 0.1% Tween-20. Clarity enhanced chemiluminescence (Bio-Rad Laboratories, Mississauga, ON) was used to visualize proteins with ChemiDoc Touch Imaging system (Bio-Rad). Protein quantification was performed using Image Lab software (Bio-Rad) by sequentially normalizing against a loading control and against vehicle-treated condition. We used rabbit polyclonal antibodies against VAPB (1:3000, HPA013144, Sigma-Aldrich) and vinculin (1:1000, 4650, Cell Signalling Technologies), rabbit XP® monoclonal antibodies against Rab7 (1:100, D95F2, Cell Signalling Technologies), mouse monoclonal antibodies against clathrin heavy chain (1:500, sc-12734, Santa Cruz Biotechnology), Arl8a/b (1:500, clone H8, Santa Cruz Biotechnology), and ATP5A (1:2000, ab14748, Abcam), rat monoclonal antibodies against LAMP1 (1:200–1:500, 1D4B, Developmental Studies Hybridoma Bank, Iowa City, IO or Santa Cruz Biotechnology), and goat polyclonal antibody against dynamin 2 (1:1000, sc-6400, Santa Cruz Biotechnology). Secondary antibodies were raised in donkey (Bethyl) and HRP-conjugated.

## Phosphoinositide labelling with $^3$H-*myo*-inositol and HPLC-coupled flow scintillation

RAW cells were incubated for 24 h with inositol-free DMEM (MP Biomedica, CA) containing 10 µCi/ml *myo*-[2-$^3$H(N)] inositol (Perkin Elmer, MA), 1X insulin-transferrin-selenium-ethanolamine (Gibco), 10% dialyzed FBS (Gibco), 4 mM L-glutamine (Sigma-Aldrich) and 20 mM HEPES (Gibco). Cells were then treated with rotenone, $H_2O_2$ and/or apilimod as indicated.

Cells were lysed and lipids precipitated with 600 μl of 4.5% perchloric acid (v/v) for 15 min on ice, collected by scraping and pellet obtained at 12000 x*g* for 10 min. Then, 1 ml of 0.1 M EDTA was used to wash pellets followed by resuspension in 50 μl water. This was followed by 500 μl of methanol/40% methylamine/1-butanol [45.7% methanol: 10.7% methylamine: 11.4% 1-butanol (v/v)] used for 50 min at 53˚C to deacylate phospholipids. Sample pellets were vaccum-dried and washed twice in 300 μl water with vaccum-drying. Deacylated phospholipids were extracted from dried sample pellets by resuspending pellet in 450 μl water and 300 μl 1-butanol/ethyl ether/ethyl formate (20:4:1), vortexing 5 min, followed by centrifugation 12000 x*g* for 2 min and then the bottom aqueous layer was collected. Extraction was performed three times followed by vaccum-drying the aqueous layer and resuspending lipids in 50 μl water. For all treatment samples, equal $^3$H counts were loaded and separated by HPLC (Agilent Technologies, Mississauga, ON) through 4.6 x 250-mm anion exchange column (Phenomenex, Torrance, CA) using a 1 ml/min flow rate with a gradient set with water (buffer A) and 1 M $(NH_4)_2HPO_4$, pH 3.8 (phosphoric acid adjusted) (buffer B) as follows: 0% B for 5 min, 0 to 2% B for 15 minutes, 2% B for 80 minutes, 2 to 10% B for 20 minutes, 10% B for 30 minutes, 10 to 80% B for 10 minutes, 80% B for 5 minutes, 80 to 0% B for 5 minutes. Radiolabel signal was detected with a 1:2 ratio of eluate to scintillant (LabLogic, Brandon, FL) in a β-RAM 4 (LabLogic) and analyzed by Laura 4 software. Each phosphoinositide species detected was normalized against the parent phosphatidylinositol peak as described in [48].

## Phagocytosis particle preparation and phagosome maturation assays

pZsGreen-containing bacteria were grown at 37˚C in Lysogeny Broth (LB), supplemented with 1% glucose to suppress leaky ZsGreen expression, and 100 μg/mL ampicillin (LB Growth Media). To produce ZsGreen-expressing bacteria, bacteria cultures were grown overnight in liquid LB Growth Media. The bacteria culture was then subcultured 1:100 in LB supplemented with ampicillin and without glucose (LB Expression Media) and incubated at 37˚C to mid-log growth phase. Isopropylthio-β-galactoside was added into the subculture to a final concentration of 100 μM, and the subculture was incubated for another 3 hours. Bacteria were washed with PBS, then fixed with 4% PFA, and stored at 4˚C in PFA. Prior to use, fixed bacteria were washed with PBS to remove PFA.

RAW macrophages at 30 to 60% confluence were treated with 1 mM $H_2O_2$ or 0.1% dd$H_2O$ (vehicle control) for 1 h. Subsequently, $8.0 \times 10^7$ bacteria (0.1 OD × 1 mL) were introduced to macrophages and centrifuged at 400 x g for 5 minutes to synchronize phagocytosis. Macrophages were incubated for 20 minutes in the presence of $H_2O_2$ or dd$H_2O$ before washing with PBS and incubating in media containing $H_2O_2$ or dd$H_2O$ for 40 minutes. Except for PBS wash, macrophage exposure to $H_2O_2$ or vehicle was uninterrupted. Macrophages were washed with PBS then fixed with 4% PFA. Cells were then incubated in 1% w/v glycine to quench PFA. Cells were then blocked with 1% Bovine Serum Albumin (BSA), then external bacteria were immunolabeled with rabbit anti-*E. coli* antibodies (1:100, Bio-Rad Antibodies, 4329–4906), followed by DyLight 650-conjugated donkey anti-rabbit IgG antibodies (1:1000, Bethyl Laboratories, Inc., A120-208D5). Cells were then permeabilized with ice-cold methanol and blocked with 1% BSA. LAMP-1 lysosomal marker protein was immunolabeled with rat anti-LAMP-1 antibodies (1:100, 1D4B, Developmental Studies Hybridoma Bank, Iowa City, IO), followed by DyLight 550-conjugated donkey anti-rat IgG antibodies (1:1000, Bethyl Laboratories, Inc., A110-337D3). Coverslips were mounted with Dako Fluorescence Mounting Medium (Agilent, S302380-2) for imaging.

FIJI was used for image processing and quantitative image analysis of phagosome maturation. Internal bacteria masks were produced by applying a subtraction mask using external

bacteria signal. "Noise" particles defined as being a few pixels in size were removed manually. The internal bacteria mask was converted to binary and dilated to reach the edges of the phagosomes (LAMP1 signal). LAMP-1 signal colocalized to the internal bacteria mask was analyzed cell-by-cell, and the mean LAMP-1 fluorescence intensity per cell was obtained.

## Statistical analysis

All experiments were performed independently at least three times. Respective figure legends indicate number of cells/samples assessed, mean, standard error of mean (s.e.m.) and number of independent experiments. For analysing significant difference between various treatment groups, we used unpaired Student's t-test when comparing two groups only or one-way ANOVA test when comparing multiple treatment conditions in non-normalized controls. Tukey's *post hoc* test coupled to ANOVA tests was used to evaluate pairwise conditions. Statistical significance was defined as $P > 0.05$. Software used for analysis was GraphPad Prism 8.

## Supporting information

**S1 Fig. Microscopy laser induced photodamage stimulate ROS production.** (A) RAW cells were loaded with Alexa546-conjugated dextran to label lysosomes and incubated with 1 mg/mL NBT for 30 min in dark. Cells were then exposed to the red laser to excite Alexa546-conjugated dextran every 20 s or 10 s or 5 s, or not exposed to the red laser (ctrl). The NBT fluorescence was then detected using far-red channel. Scale bar: 20 μm. (B) Quantification of mean NBT fluorescence per cell. Data represent ± SEM from three independent experiments, with 20–35 cells assessed per treatment condition per experiment. One-way ANOVA and Tukey's *post-hoc* test was used, where * indicates statistical significance between indicated conditions ($p < 0.05$).
(TIF)

**S2 Fig. Lower $H_2O_2$ concentration prevents apilimod induced lysosome coalescence.** (A) RAW cells were pre-labelled with Lucifer yellow and exposed to either vehicle, 100 μM $H_2O_2$ for 40 min in presence or absence of 1 nM or 5 nM apilimod 40 min. Scale bar: 5 μm. (B-D) Quantification of individual lysosome volume (B), lysosome number per cell (C), and total lysosome volume per cell (D). Data are illustrated as mean ± SEM from three independent experiments, with 25–30 cell assessed per condition per experiment. One-way ANOVA and Tukey's *post-hoc* test was used, where * indicates $P < 0.05$ for the indicated conditions.
(TIF)

**S3 Fig. ROS prevent lysosome enlargement during acute PIKfyve suppression in HeLa and RPE cells.** (A) HeLa cells pre-labelled with Lucifer yellow and exposed to vehicle or 100 nM apilimod 40 min, or with 1 mM $H_2O_2$ in the presence or absence of 100 nM apilimod for 40 min. Scale bar: 10 μm. (B-D) Quantification of individual lysosome volume per lysosome (B), lysosome number per cell (C), and total lysosome volume per cell (D). (E) RPE cells pre-labelled with Lucifer yellow and exposed to vehicle, or 1 mM $H_2O_2$, or 10 μM CDNB, in presence or absence of 200 nM apilimod 40 min. Scale bar: 20 μm. (F-H) Quantification of individual lysosome volume (F), lysosome number per cell (G), and sum lysosome volume per cell (H). For (B-D) and (F-H), data are represented as mean ± SEM. from three independent experiments, with 25–30 cells assessed for (B-D) and 15–20 cells assessed for (F-H) per treatment condition per experiment. One-way ANOVA and Tukey's *post-hoc* test used with *$P < 0.05$ compared to indicated control conditions.
(TIF)

**S4 Fig. Quantification and validation of microtubule morphology by image analysis.** Single z-focal plane immunofluorescence micrographs of RAW cells (A) or RPE cells (E) treated with vehicle, $H_2O_2$ or rotenone. After treatment with ROS agents, cells were fixed and immunostained with anti-α-tubulin antibodies. Cells were analyzed for their microtubule morphology using the ImageJ "skeleton" plugin, converting images into binary "skeleton" micrographs. Quantification of number of microtubule junctions per cell, number of microtubule branches per cell and average microtubule branch length for RAW cells (B-D) and RPE cells (F-H). RPE cells were also analyzed for maximum microtubule patch area per cell (I) through ImageJ using binary filter and watershed segmentation. Data are represented as mean ± SD from 5 different fields of view for RAW cells or 10 different fields of view for RPE cell, with 50–70 cells assessed per treatment condition for RAW cells (A-D) and 15–20 cells assessed per treatment condition for RPE cells (E-H). One-way ANOVA and Tukey's *post-hoc* test used for B-D and F-H, where * indicates statistically significant difference between control conditions ($P<0.05$). Scale bar: 20 μm (A, E). (TIF)

**S5 Fig. Increased microtubule stability does not affect lysosome motility or lysosome coalescence during PIKfyve inhibition.** (A) RAW cells pre-labelled with Lucifer yellow were exposed to either vehicle, or 1 μM or 10 μM paclitaxel for 60 min in presence or absence of 20 nM apilimod for the remaining 40 min. Scale bar: 5 μm. (B-D) Quantification of individual lysosome volume (B), lysosome number per cell (C), and total lysosome volume per cell (D). Data are represented as mean ± s.e.m. from three independent experiments, with 25–30 cell assessed for (B-D) per treatment condition per experiment. (E-G) RAW cells pre-labelled with Lucifer yellow were exposed to vehicle or 1 μM or 10 μM paclitaxel 60 min. Live cell spinning disc confocal microscopy was performed at single z-focal plane once every 4 sec for 3 min. Quantification of lysosome speed (E), lysosome displacement (F), and lysosome track length (G) are shown. Data are represented as mean ± s.d. from three independent experiments. One-way ANOVA and Tukey's *post-hoc* tests were used, where * indicates $P<0.05$ between experimental and control conditions. Data is based on movies like those represented by S14–S16 Movies. (TIF)

**S6 Fig. ROS do not affect Rab7 activation and Arl8b loading onto lysosomes.** RAW cells expressing RILPC33-GFP (A), or Arl8bWT-GFP (B), exposed to vehicle in absence or presence of 20 nM apilimod 40 min, or 1 mM $H_2O_2$ 40 min in presence or absence of 20 nM apilimod 40 min. Scale bar: 5 μm. (C-D) Quantification of membrane associated fluorescence intensity of RILPC33-GFP (C) from (A) or Arl8bWT-GFP (D) from (B), normalized to cytosol fluorescence intensity. Data represent mean ± SEM from three independent experiments, with 15–20 cell assessed per treatment condition per experiment. One-way ANOVA and Tukey's *post-hoc* test used for C-D with *$P<0.05$ compared to indicated control conditions. (E) A representative Western blot of membrane fractions from RAW macrophages treated with vehicle, rotenone, or $H_2O_2$ with or without apilimod. Blots were probed with antibodies against Rab7, Arl8a/b, and LAMP1, the latter used to benchmark membrane levels. (F) Relative levels of Arl8ab/b or Rab7 as a ratio to LAMP1 band intensity. Data are shown as mean + standard deviation from n = 3 independent experiments. (TIF)

**S7 Fig. $H_2O_2$ boosts recruitment of clathrin and dynamin to membranes.** (A) RPE cells stably expressing clathrin heavy chain-eGFP were pre-labelled with Alexa[546-]conjugated dextran and treated with vehicle, 1 mM $H_2O_2$, or 200 nM apilimod with or without 1 mM $H_2O_2$. Single

z- plane images were acquired every 2 min for 40 min across all treatments. Fluorescence micrographs represent single z-plane images at 0 min and 40 min for each treatment obtained by spinning disc microscopy. The inset is a magnified portion of field of view tracking Alexa[546]-conjugated dextran lysosome(s) or clathrin-eGFP separate or merged. Scale bar: 7 µm. B. Ratio of clathrin-eGFP fluorescence intensities associated with Alexa[546]-conjugated dextran and cytosol time points: 0, 10, 20, 30, and 40 min. Data are represented as mean ± s.e. m. from five to six independent experiments, with 1–3 cells assessed per treatment condition per experiment. Two-way ANOVA and Tukey's *post-hoc* test were used for (B), where * indicates $P<0.05$ against control conditions. (C) RAW cells were treated with vehicle or 1 mM $H_2O_2$ for 40 min, lysed and homogenates fractionated through a sucrose gradient ultracentrifugation. Fractions were immunoblotted against LAMP1 and VAPB to respectively identify lysosome and ER fractions, and aganst clathrin heavy chain and dynamin 2. Protein expression for clathrin heavy chain (D) or dynamin 2 (E) were normalized to LAMP1 for fractions 3 to 6. Data are represented as mean ± s.d. from three independent experiments. Unpaired Student's t-test was used for (D-E), where * indicates $P<0.05$ against vehicle control conditions. (TIF)

**S8 Fig. Dynamin inhibition does not affect lysosome fragmentation during during PIKfyve reactivation.** (A) RAW cells were pre-labelled with Lucifer yellow and exposed to either vehicle, 30 µM dyngo-4A for 2 h, 1 mM $H_2O_2$ for 40 min, or 1 µM rotenone for 1 h, or 20 nM apilimod for 60 min. Additional subgroup of apilimod treated cells were then washed and incubated with apilimod-free media and changed for 2 h in the presence of vehicle, dyngo-4A, H2O2, and dyngo4-A plus H2O2 for a total time of 2 h without apilimod. Fluorescence micrographs are spinning disc microscopy images with 45–55 z-planes represented as z-projections. Scale bar: 5 µm. (B-D) Quantification of individual lysosome volume (B), lysosome number per cell (C), and total lysosome volume per cell (D). Data is illustrated as mean ± s.e.m. from three independent experiments, with 25–30 cell assessed per treatment condition per experiment. One-way ANOVA and Tukey's *post-hoc* test used for B-D with *$P<0.05$ compared to indicated control conditions. (TIF)

**S1 Raw images. Original Western blots displayed in this manuscript that are uncropped and untouched.** (PDF)

**S1 Movie. Lysosome motility for vehicle-treated RAW macrophages.** Live-cell imaging of RAW macrophages pre-labelled with Lucifer yellow and treated with vehicle-only. Single-plane acquired every 4 sec for 3 min. Time and scale are as indicated. (MP4)

**S2 Movie. Lysosome motility for H2O2-treated RAW macrophages.** Live-cell imaging of RAW macrophages pre-labelled with Lucifer yellow and treated with 1 mM H2O2 for 40 min. Single-plane acquired every 4 sec for 3 min. Time and scale are as indicated. (MP4)

**S3 Movie. Lysosome motility for rotenone-treated RAW macrophages.** Live-cell imaging of RAW macrophages pre-labelled with Lucifer yellow and treated with 1 µM rotenone for 60 min. Single-plane acquired every 4 sec for 3 min. Time and scale are as indicated. (MP4)

**S4 Movie. Lysosome motility for CDNB-treated RAW macrophages.** Live-cell imaging of RAW macrophages pre-labelled with Lucifer yellow and treated with 10 µM CDNB 30 min.

Single-plane acquired every 4 sec for 3 min. Time and scale are as indicated.
(MP4)

**S5 Movie. Lysosome motility for MCB-treated RAW macrophages.** Live-cell imaging of
RAW macrophages pre-labelled with Lucifer yellow and treated with 5 μM MCB 30 min. Single-plane acquired every 4 sec for 3 min. Time and scale are as indicated.
(MP4)

**S6 Movie. Lysosome motility for nocodazole-treated RAW macrophages.** Live-cell imaging
of RAW macrophages pre-labelled with Lucifer yellow and treated with 10 μM nocodazole 60
min. Single-plane acquired every 4 sec for 3 min. Time and scale are as indicated.
(MP4)

**S7 Movie. Lysosome motility for vehicle-treated RPE cells.** Live-cell imaging of RPE cells
pre-labelled with Lucifer yellow and treated with vehicle-only. Single-plane acquired every 8
sec for 6 min. Time and scale are as indicated.
(MP4)

**S8 Movie. Lysosome motility for H$_2$O$_2$-treated RPE cells.** Live-cell imaging of RPE cells pre-labelled with Lucifer yellow and treated with 1 mM H$_2$O$_2$ 40 min. Single-plane acquired every
8 sec for 6 min. Time and scale are as indicated.
(MP4)

**S9 Movie. Lysosome motility for rotenone-treated RPE cells.** Live-cell imaging of RPE cells
pre-labelled with Lucifer yellow and treated with 1 μM rotenone 60 min. Single-plane acquired
every 8 sec for 6 min. Time and scale are as indicated.
(MP4)

**S10 Movie. Lysosome motility for CDNB-treated RPE cells.** Live-cell imaging of RPE cells
pre-labelled with Lucifer yellow and treated with 10 μM CDNB 30 min. Single-plane acquired
every 8 sec for 6 min. Time and scale are as indicated.
(MP4)

**S11 Movie. Lysosome motility for MCB-treated RPE cells.** Live-cell imaging of RPE cells
pre-labelled with Lucifer yellow and treated with 5 μM MCB 30 min. Single-plane acquired
every 8 sec for 6 min. Time and scale are as indicated.
(MP4)

**S12 Movie. Lysosome motility for nocodazole five micromolar treated RPE cells.** Live-cell
imaging of RPE cells pre-labelled with Lucifer yellow and treated with 5 μM nocodazole 60
min. Single-plane acquired every 8 sec for 6 min. Time and scale are as indicated.
(AVI)

**S13 Movie. Lysosome motility for nocodazole ten micromolar treated RPE cells.** Live-cell
imaging of RPE cells pre-labelled with Lucifer yellow and treated with 10 μM nocodazole 60
min. Single-plane acquired every 8 sec for 6 min. Time and scale are as indicated.
(AVI)

**S14 Movie. Lysosome motility for vehicle-treated RAW macrophages.** Live-cell imaging of
RAW cells pre-labelled with Lucifer yellow and treated with vehicle-only. Single-plane
acquired every 4 sec for 3 min. Time and scale are as indicated.
(MP4)

**S15 Movie. Lysosome motility for paclitaxel one micromolar treated RAW macrophages.**
Live-cell imaging of RAW cells pre-labelled with Lucifer yellow and treated with 1 µM pacli-
taxel 60 min. Single-plane acquired every 4 sec for 3 min. Time and scale are as indicated.
(MP4)

**S16 Movie. Lysosome motility for paclitaxel ten micromolar treated RAW macrophages.**
Live-cell imaging of RAW cells pre-labelled with Lucifer yellow and treated with 10 µM pacli-
taxel 60 min. Single-plane acquired every 4 sec for 3 min. Time and scale are as indicated.
(MP4)

## Acknowledgments

ARPE-1 (RPE) cells stably expressing clathrin heavy chain-eGFP were a kind gift from Dr.
Costin Antonescu. We would also like to thank Mr. Janusan Baskararajah and Mr. Nemanja
Ilic for assistance in data analysis.

## Author Contributions

**Conceptualization:** Golam T. Saffi, Roberto J. Botelho.

**Data curation:** Golam T. Saffi.

**Formal analysis:** Golam T. Saffi, Evan Tang, Sami Mamand, Subothan Inpanathan, Aaron
Fountain.

**Funding acquisition:** Roberto J. Botelho.

**Investigation:** Golam T. Saffi, Evan Tang, Sami Mamand, Subothan Inpanathan, Aaron Foun-
tain, Roberto J. Botelho.

**Methodology:** Golam T. Saffi, Evan Tang, Sami Mamand, Subothan Inpanathan, Aaron
Fountain.

**Project administration:** Roberto J. Botelho.

**Resources:** Leonardo Salmena, Roberto J. Botelho.

**Software:** Golam T. Saffi.

**Supervision:** Leonardo Salmena, Roberto J. Botelho.

**Visualization:** Golam T. Saffi, Evan Tang, Sami Mamand, Subothan Inpanathan, Aaron
Fountain.

**Writing – original draft:** Golam T. Saffi, Roberto J. Botelho.

**Writing – review & editing:** Golam T. Saffi, Leonardo Salmena, Roberto J. Botelho.

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
