## [Decision Letter · Decision Letter 0]

20 Aug 2021

PONE-D-21-19956

Reactive oxygen species prevent lysosome coalescence during PIKfyve inhibition

PLOS ONE

Dear Dr. Botelho,

Thank you for submitting your manuscript to PLOS ONE. After careful consideration, we feel that it has merit but does not fully meet PLOS ONE’s publication criteria as it currently stands. Therefore, we invite you to submit a revised version of the manuscript that addresses the points raised during the review process.

We look forward to receiving your revised manuscript.

Kind regards,

David Chau

Academic Editor

PLOS ONE

Journal Requirements:

Reviewers' comments:

Reviewer's Responses to Questions

**Comments to the Author**

1. Is the manuscript technically sound, and do the data support the conclusions?

Reviewer #1: Partly

Reviewer #2: No

Reviewer #3: Yes

2. Has the statistical analysis been performed appropriately and rigorously? 

Reviewer #1: Yes

Reviewer #2: No

Reviewer #3: Yes

3. Have the authors made all data underlying the findings in their manuscript fully available?

Reviewer #1: Yes

Reviewer #2: Yes

Reviewer #3: Yes

4. Is the manuscript presented in an intelligible fashion and written in standard English?

Reviewer #1: Yes

Reviewer #2: Yes

Reviewer #3: Yes

5. Review Comments to the Author

Reviewer #1: This is an extensive examination of ROS contribution to lysosome with many supportive evidence. Kudos to the authors. I realize that I was the forth reviewer for the manuscript, which is in fact a revised one. I have some concerns similar to reviewer 3. CDNB is not a Trx inhibitor and is more of thiol inhibitor and should be clearly stated in the text. PX12 would be my preference for Trx but since Auranofin is used for only 2 hours I would accept that. I didn't see any data on cell survival. Does this dose affect cell survival?

The question for Lysosomal activity remains. Why not simply using lysotracker staining and quantifying the lamp-positive particle containing lysotracker. There are good references on lysosome activity and size I found (a simple Pubmed search) including 27693380, 28478025. These are relevant recent works that should be cited. The same group also looks at the changes in lysosomal size and shows the effect as size distribution. I don't think showing the results as overall number or volume quite captures the effect that authors are trying to show. This will add to the impact of the paper.

Please check the abbreviation.

Reviewer #2: The authors observed that photo-damage during live-cell imaging prevented lysosome fusion during PIKfyve inhibition. They postulated that lysosome fusion and/or fission dynamics are affected by reactive oxygen species (ROS). Here they present evidence that ROS prevent lysosome fusion in the presence of PIKfyve phosphoinositide kinase inhibitors.

The major concerns in this review are that the data presented do not support their hypothesis, and that the conclusions presented are contradictory to at least two previous publications that are not mentioned in this report. Therefore, we cannot recommend publication.

Major Comments:

1. The authors hypothesize that that excessive exposure of cells to the laser beam during confocal microscopy generates ROS that inhibits lysosome size enlargement under acute PIKFYVE inhibition. However, the authors did not perform ROS measurements from cells after laser exposure in order it to confirm their conclusions.

2. The authors concluded that ROS accelerate the reversal of lysosome enlargement during Apilimod washout (Figure 4). However, comparing the of lysosome size by ROS in the presence (Figure 2B) and in the absence (Figure 4B) of Apilimod does not confirm that conclusion, because lysosome size under both conditions was reduced between 2 to 2.5 fold. Therefore, the statement in Line 179 “Overall, ROS prevented and reversed lysosome coalescence induced by PIKfyve inhibition” is not supported by the experimental results presented in this manuscript.

3. It is well known that inhibition of autophagy by different methods (including the use of PIKFYVE inhibitors) increases ROS production in cells. The authors need to quantify the ROS level during treatment with Apilimod alone and then compare it with different ROS agonists. Comparing the ROS level with and without the apilimod in the presence of ROS agonists would strengthen the conclusion of this finding.

4. The phosphoinositide, PI(4,5)P2 known to cause lysosome reformation. What is the effect of co-exposure of cells to apilimod and ROS agonists on PI(4,5)P2 levels? Quantifying the level of PI(4, 5)P2 along with PI3P and PI(3,5)P2 would prove that ROS plays a key role in preventing lysosome enlargement during PIKfyve inhibition.

5. What is the co-exposure of cells to apilimod and ROS agonists on autophagy inhibition in cells? Is it more or less than autophagy inhibition by Apilimod alone?

6. The authors carried out many experiments to delineate various pathways for the reduction in lysosome volume upon exposure to ROS agonists to cells with acute PIKfyve inhibition. However, the functions of proteins such as the vacuolar H+-ATPase (V-ATPase), the cholesterol transporter, and the lysosome membrane fusion complex whose inhibition is known to inhibit lysosome enlargement during Pikfyve inhibition are not discussed in this manuscript. The authors should explore the inhibition of these molecules in PIKfyve inhibited cells after ROS agonist treatment.

7. This study contradicts the findings of an earlier study that concluded that PIKfyve inhibitors caused prolonged presence of the NADPH oxidase NOX2, which resulted in increased reactive oxygen species (ROS) production (Dingjan et al., 2017). More recently, Baranov et al., 2019, showed that pharmacological inhibition of PIKfyve promoted NOX2-mediated ROS production in dendritic cells. They further confirmed that the treatment of dendritic cells with apilimod or YM201636 resulted in massive cellular vacuolization, an establshed effect of PIKfyve inhibition. In contrast, the present study by Saffi and co-workers conclude that ROS prevents lysosome fusion during PIKfyve inhibition. In their study, treating the cells with ROS inducers following PIKfyve inhibitor treatment, prevented lysosome enlargement. This observation contradicts the observations reported by Baranove et al. where PIKfyve inhibited cells exhibited lysosome enlargement and produced ROS. The authors should explain this discrepancy between the studies. In order to determine the status of ROS production following PIKfyve inhibition, the authors should also measure the level of ROS in PIKfyve inhibitor treated cells.

Neither of these studies was mentioned in the Saffi et al., manuscript submitted here.

Dingjan et al., J Cell Sci. 2017, PMID: 28202687

Baranov et al., iScience. 2019, PMID: 30612035

Minor Comment

1. In Figure 1A, “CDNB (µM)” should read “CDNB”.

Reviewer #3: Saffi and co-workers have studied the underlying mechanisms of ROS impairing lysosome coalescence, as acutely induced by PIKfyve inhibition using apilimod. Their study originated from their serendipitous finding that extended spinning disk confocal analysis impacted on lysosomal coalescence induced by PIKfyve inhibition, arguing for the involvement of ROS. Having strong expertise in lysosomal biogenesis/dynamics and phosphoinositide signaling, the authors performed a profound and systematic analysis of distinct possible mechanisms, ruling out for instance a role for increased PI3,5P. They find effects on microtubule stability but they remained disparate for different ROS species, arguing against a primary role. They found that H2O2 imparied the ability of lysosomes to fuse with target organelles, but this was not through increased membrane disruption or altered calcium release. No relation was found either with an altered recruitment of fission machineries such as clathrin and dynamin, ultimately leading to a role of ROS in actin depolymerization thereby preventing lysosomal coalescence.

The amount of data is impressive, of high quality and the fact that they were performed in such a detailed and systematic manner, supports publication. This is as well a revised manuscript, and although I did not review it in the first round, the way the authors addressed all critiques is highly appreciated and clearly further improved the quality of the data and interpretations.

6. PLOS authors have the option to publish the peer review history of their article (what does this mean?). If published, this will include your full peer review and any attached files.

Reviewer #1: No

Reviewer #2: No

Reviewer #3: No

---

## [Author Response · Author response to Decision Letter 0]

25 Sep 2021

Response to Reviewers

We thank the reviewers for the additional comments and opportunity to further strengthen our investigative work. The detailed rebuttal and inventory of changes done to the manuscript are recorded below.

Reviewer #1: 

R1.1: This is an extensive examination of ROS contribution to lysosome with many supportive evidence. Kudos to the authors. I realize that I was the forth (sic) reviewer for the manuscript, which is in fact a revised one. I have some concerns similar to reviewer 3. CDNB is not a Trx inhibitor and is more of thiol inhibitor and should be clearly stated in the text. PX12 would be my preference for Trx but since Auranofin is used for only 2 hours I would accept that. I didn't see any data on cell survival. Does this dose affect cell survival?

RESPONSE: Thank you for the positive comments on our work. To address the raised concerns, we now refer to CDNB as a “thiol inhibitor” and to Auranofin as a Trx inhibitor. This can be found throughout the manuscript and illustrated with lines 143 and 164, 437. In addition, we measured survival in cells treated with select treatments [(control vs apilimod) and (H2O2, rotenone, CDNB, and auranofin with vs. without apilimod)] using propidium iodide exclusion assay. These treatments are relatively acute (less than 2 h); while more than enough time to impact lysosome enlargement, we did not observe significant change in PI staining, indicating cell survival was not affected within our experimental parameters. This is shown in Figure 1I and indicated in lines 154.

R1.2: The question for Lysosomal activity remains. Why not simply using lysotracker staining and quantifying the lamp-positive particle containing lysotracker. There are good references on lysosome activity and size I found (a simple Pubmed search) including 27693380, 28478025. These are relevant recent works that should be cited. The same group also looks at the changes in lysosomal size and shows the effect as size distribution. I don't think showing the results as overall number or volume quite captures the effect that authors are trying to show. This will add to the impact of the paper.

RESPONSE: We previously had measured lysosome proteolysis using Magic Red but did not include it in the manuscript. We now include these data in a new Figure 10A-C. We measured lysosome pH using Lysotracker, which is also in new Figure 10D, E. We did not observe a significant change in LysoTracker intensity under any of these conditions (Figure 10E). As for degradation, we used Magic Red Cathepsin substrate, a fluorogenic probe that measures Cathepsin B activity, whereby increased fluorescence indicates enhanced degradation. To account for differences in loading or due to changes in lysosome size, we normalized fluorescence of Magic Red to Alexa488-conjugated dextran fluorescence pre-loaded onto lysosomes. We then measured the ratio of Magic Red to Alexa488-conjugated dextran per cell in resting cells vs. apilimod-treated cells. However, within the time frame of 1 h, ROS agents alone or apilimod alone did not significantly alter proteolytic activity (Fig. 10C). However, our experiments do reveal an interaction by combining apilimod and ROS, which led to significant reduction in Magic Red signal relative to cells treated with the respective ROS agent alone (eg. compare H2O2 alone vs. apilimod+H2O2). Thus, despite offsetting lysosome enlargement, there is a tendency for ROS to reduce proteolytic activity when combined with apilimod. This is indicated in lines 324-333.

It is possible that longer periods of incubation with apilimod or ROS agents might lead to greater impact on proteolysis or pH, but these would then be indirect effects in our opinion, which we are trying to limit. We cite the above works (27693380 and 28478025) in the Introduction (lines 119), in the Results section (Lines 304, 318). However, we note that cells were stressed for >6 h or overnight in these studies and thus represent distinct conditions, which we highlight in line 318. These are important findings that speak to the repercussions of ROS on lysosome and autophagy, but again, we are trying to limit our observations to direct effects of PIKfyve inhibition and ROS on lysosome size. 

R1.3: Please check the abbreviation.

RESPONSE: We added missing abbreviations like BAPTA-AM, DMEM; FYCO1, PIKfyve, LC3.

REVIEWER #2

We thank Reviewer 2 for the additional comments and critical assessment of our work. We believe we have addressed the concerns appropriately.

R2: The major concerns in this review are that the data presented do not support their hypothesis, and that the conclusions presented are contradictory to at least two previous publications that are not mentioned in this report. Therefore, we cannot recommend publication.

RESPONSE: We agree with the Reviewer that we need to acknowledge seemingly contradictory works and discuss the context between our work and other relevant publications. We apologize for omitting their citation and for not discussing relevant works here, which we now addressed and detailed below. However, we do hope that the Reviewer(s) agree that contradictory observations on their own are not grounds to not recommend publication, as long as data are controlled, and this is acknowledged and discussed. 

R2.1: The authors hypothesize that that excessive exposure of cells to the laser beam during confocal microscopy generates ROS that inhibits lysosome size enlargement under acute PIKFYVE inhibition. However, the authors did not perform ROS measurements from cells after laser exposure in order it to confirm their conclusions.

RESPONSE: We had done these experiments before but did not include them. Thank you for raising the issue. We show in New S1 Fig and indicate in line 138 that RAW cells increasingly exposed to laser light have significantly more ROS as detected by NBT staining. 

R2.2: The authors concluded that ROS accelerate the reversal of lysosome enlargement during Apilimod washout (Figure 4). However, comparing the of lysosome size by ROS in the presence (Figure 2B) and in the absence (Figure 4B) of Apilimod does not confirm that conclusion, because lysosome size under both conditions was reduced between 2 to 2.5 fold. Therefore, the statement in Line 179 “Overall, ROS prevented and reversed lysosome coalescence induced by PIKfyve inhibition” is not supported by the experimental results presented in this manuscript.

RESPONSE: If the reviewer is referring to the difference in the absolute “average lysosome volume” (the size of individual lysosomes) between Figures 2B and Figure 4B, we would argue that this is not the right comparison because the average diameter of individual lysosome induced by apilimod can change between experiments done at different times over the lifetime of this project – for example, after apilimod treatment, the average diameter of individual lysosomes has been recorded as 6 µm in Fig. 2B, 9 µm in Fig. 10D, 10 µm in Fig. 11B, 12 µm in Fig. 12D. So, absolute average size can change between experiments done over a couple of years. 

The lysosome size for each figure is consistent because we clustered our independent experiments for each figure. Several reasons, such as variation in the age of the drug and/or cell passage, could cause the absolute average diameter of the lysosome to be different between experiments conducted over a couple of years. Overall, our comparisons are between data and conditions within an experiment, not between figures. 

Regardless of this, adding apilimod to cells causes the average diameter of individual lysosomes to increase – if we co-administer apilimod and ROS agents, the increase is less prominent, suggesting that ROS help prevent lysosome enlargement during PIKfyve inhibition (Fig. 2). On the other hand, if we add apilimod to cause lysosome enlargement and then we wash the drug and chase with just fresh media, lysosomes begin to shrink in size. The key comparison is that if we administer ROS agents during this chase-period, lysosomes get even smaller relative to cells with fresh media-only during the chase (Fig. 4). So, since adding ROS elicits lysosomes to become smaller (and more numerous) relative to fresh media-only during the chase to reverse lysosome enlargement caused by apilimod, we conclude that ROS help reverse the coalescence caused by PIKfyve inhibition. 

We concede that “accelerate” is not the best word since we are not presenting kinetics. Perhaps this was the issue. If so, we replaced “accelerate” with “helps to reverse” or “promotes” in the abstract, conclusion statement, and in the main and relevant section of Results related to Figure 4, starting in lines 187.

R2.3: It is well known that inhibition of autophagy by different methods (including the use of PIKFYVE inhibitors) increases ROS production in cells. The authors need to quantify the ROS level during treatment with Apilimod alone and then compare it with different ROS agonists. Comparing the ROS level with and without the apilimod in the presence of ROS agonists would strengthen the conclusion of this finding.

RESPONSE: We exposed cells to apilimod (~1 h, 20 nM) with and without rotenone or H2O2 and then incubated with CellRox Green to quantify ROS. While rotenone and H2O2 increase the CellRox Green signal, cells exposed to apilimod-alone did not significantly have more CellRox Green signal relative to vehicle alone cells. Moreover, apilimod-treatment did not increase the level of ROS caused by rotenone or H2O2. These data are presented in Figure 9E, 9F and indicated in lines 319. Hence, there is no apparent increase in ROS under acute apilimod exposure, conditions sufficient to alter lysosome coalescence. On the other hand, studies like those by Baranov et al. show an increase in ROS during PIKfyve inhibition, but we note that they used µM range of YM201636 for 3 h. Thus, the effect of PIKfyve inhibition on ROS may require prolonged inhibition or depend on other factors like the cell type. We discuss this in lines 322, 415, and 466.

R2.4: The phosphoinositide, PI(4,5)P2 known to cause lysosome reformation. What is the effect of co-exposure of cells to apilimod and ROS agonists on PI(4,5)P2 levels? Quantifying the level of PI(4, 5)P2 along with PI3P and PI(3,5)P2 would prove that ROS plays a key role in preventing lysosome enlargement during PIKfyve inhibition.

RESPONSE: As shown in the revised Figure 5E, 5F and indicated in lines 231, we observed no changes to PI(4,5)P2 in cells with or without apilimod, H2O2, or rotenone. 

R2.5: What is the co-exposure of cells to apilimod and ROS agonists on autophagy inhibition in cells? Is it more or less than autophagy inhibition by Apilimod alone?

RESPONSE: We expressed the autophagy and autophagosome flux biosensor, LC3-GFP-mCherry. Within the time frame of < 2 h of apilimod and ROS agent exposure, which is more than enough time to alter lysosome coalescence, we did not see much of an effect on LC3-GFP puncta or the ratio of GFP/mCherry – this is indicated in Figure 9C, D and E and mentioned in line 313. While we believe that longer periods of time would impact autophagy and alter autophagasome dynamics, as others have observed, this would not explain the impact of PIKfyve/ROS on lysosome coalescence. 

R2.5: The authors carried out many experiments to delineate various pathways for the reduction in lysosome volume upon exposure to ROS agonists to cells with acute PIKfyve inhibition. However, the functions of proteins such as the vacuolar H+-ATPase (V-ATPase), the cholesterol transporter, and the lysosome membrane fusion complex whose inhibition is known to inhibit lysosome enlargement during Pikfyve inhibition are not discussed in this manuscript. The authors should explore the inhibition of these molecules in PIKfyve inhibited cells after ROS agonist treatment.

RESPONSE: We examined lysosomal proteolysis using Magic Red as an indicator of cathepsin activity and lysosomal pH using Lysotracker. We copied and pasted the response to Reviewer 1 as it addresses a similar point. 

“We previously had measured lysosome proteolysis using Magic Red but did not include it in the manuscript. We now include these data in a new Figure 10A-C. We measured lysosome pH using Lysotracker, which is also in new Figure 10D, E. We did not observe a significant change in LysoTracker intensity under any of these conditions (Figure 10E). As for degradation, we used Magic Red Cathepsin substrate, a fluorogenic probe that measures Cathepsin B activity, whereby increased fluorescence indicates enhanced degradation. To account for differences in loading or due to changes in lysosome size, we normalized fluorescence of Magic Red to Alexa488-conjugated dextran fluorescence pre-loaded onto lysosomes. We then measured the ratio of Magic Red to Alexa488-conjugated dextran per cell in resting cells vs. apilimod-treated cells. However, within the time frame of 1 h, ROS agents alone or apilimod alone did not significantly alter proteolytic activity (Fig. 10C). However, our experiments do reveal an interaction by combining apilimod and ROS, which led to significant reduction in Magic Red signal relative to cells treated with the respective ROS agent alone (eg. compare H2O2 alone vs. apilimod+H2O2). Thus, despite offsetting lysosome enlargement, there is a tendency for ROS to reduce proteolytic activity when combined with apilimod. This is indicated in lines 326-333. It is possible that longer periods of incubation with apilimod or ROS agents might lead to greater impact on proteolysis or pH, but these would then be indirect effects in our opinion, which we are trying to limit.”

R2.6: This study contradicts the findings of an earlier study that concluded that PIKfyve inhibitors caused prolonged presence of the NADPH oxidase NOX2, which resulted in increased reactive oxygen species (ROS) production (Dingjan et al., 2017). More recently, Baranov et al., 2019, showed that pharmacological inhibition of PIKfyve promoted NOX2-mediated ROS production in dendritic cells. They further confirmed that the treatment of dendritic cells with apilimod or YM201636 resulted in massive cellular vacuolization, an established (sic) effect of PIKfyve inhibition. In contrast, the present study by Saffi and co-workers conclude that ROS prevents lysosome fusion during PIKfyve inhibition. In their study, treating the cells with ROS inducers following PIKfyve inhibitor treatment, prevented lysosome enlargement. This observation contradicts the observations reported by Baranove (sic) et al. where PIKfyve inhibited cells exhibited lysosome enlargement and produced ROS. The authors should explain this discrepancy between the studies. In order to determine the status of ROS production following PIKfyve inhibition, the authors should also measure the level of ROS in PIKfyve inhibitor treated cells. Neither of these studies was mentioned in the Saffi et al., manuscript submitted here. Dingjan et al., J Cell Sci. 2017, PMID: 28202687; Baranov et al., iScience. 2019, PMID: 30612035

RESPONSE: Thank you for identifying Baranov et al., which is relevant for us to cite and discuss, which we do in lines 322, 415, and 466. However, Dingjuan et al., 2017 did not inhibit PIKfyve and instead looked at the role of lysosomal SNAREs in phagosome-delivery and recycling of Nox2 subunits. They do co-stain Nox2 subunits with a fluorescent probe for 3-PIPs but they do not functionally test the role of PI(3,5)P2 in this work. We think this work is more peripheral to our findings. Regardless of this, we do not agree that our work is necessarily contradictory to the studies in question because: 

i) Baranov et al inhibited PIKfyve and then measure ROS – they are not adding exogenous ROS or exacerbating ROS using agents like rotenone. 

ii) As we state in our discussion, different ROS type, sources, location and level may have different effects on lysosomes. Thus, the ROS formed during PIKfyve inhibition may not have an apparent effect on lysosome dynamics because the type, location and levels may be distinct from adding ROS exogenously like H2O2 or by mitochondrial decoupling with rotenone. This is now stated in Discussion, lines 464.

iii) We did not observe ROS using CellRox Green in apilimod-treated cells under acute PIKfyve inhibition, which causes lysosome enlargement - again, this is not necessarily contradictory to other studies since we are using acute PIKfyve inhibition to understand the dynamics on lysosome fusion-fission, while Baranov et al. used YM201636 at 3 µM for 3 h. This is also done in different cell types.

iv) We are not claiming that all ROS inhibit lysosome fusion. While H2O2 may impact fusion, the other ROS agents may instead stimulate fission by releasing a dense actin structure that accumulate on lysosomes during PIKfyve inhibition. 

Minor Comment

R2.7: In Figure 1A, “CDNB (µM)” should read “CDNB”.

 RESPONSE: Corrected.

Reviewer #3: 

R3: The amount of data is impressive, of high quality and the fact that they were performed in such a detailed and systematic manner, supports publication. This is as well a revised manuscript, and although I did not review it in the first round, the way the authors addressed all critiques is highly appreciated and clearly further improved the quality of the data and interpretations.

RESPONSE: We appreciate the positive commentary and view.

---

## [Decision Letter · Decision Letter 1]

12 Oct 2021

PONE-D-21-19956R1Reactive oxygen species prevent lysosome coalescence during PIKfyve inhibitionPLOS ONE

Dear Dr. Botelho,

Thank you for submitting your manuscript to PLOS ONE. After careful consideration, we feel that it has merit but does not fully meet PLOS ONE’s publication criteria as it currently stands. Therefore, we invite you to submit a revised version of the manuscript that addresses the points raised during the review process.

We look forward to receiving your revised manuscript.

Kind regards,

David Chau

Academic Editor

PLOS ONE

Journal Requirements:

Reviewers' comments:

Reviewer's Responses to Questions

**Comments to the Author**

1. If the authors have adequately addressed your comments raised in a previous round of review and you feel that this manuscript is now acceptable for publication, you may indicate that here to bypass the “Comments to the Author” section, enter your conflict of interest statement in the “Confidential to Editor” section, and submit your "Accept" recommendation.

Reviewer #1: All comments have been addressed

Reviewer #2: All comments have been addressed

Reviewer #3: All comments have been addressed

2. Is the manuscript technically sound, and do the data support the conclusions?

Reviewer #1: Yes

Reviewer #2: Yes

Reviewer #3: Yes

3. Has the statistical analysis been performed appropriately and rigorously? 

Reviewer #1: Yes

Reviewer #2: Yes

Reviewer #3: Yes

4. Have the authors made all data underlying the findings in their manuscript fully available?

Reviewer #1: Yes

Reviewer #2: Yes

Reviewer #3: Yes

5. Is the manuscript presented in an intelligible fashion and written in standard English?

Reviewer #1: Yes

Reviewer #2: Yes

Reviewer #3: Yes

6. Review Comments to the Author

Reviewer #1: The reviewers have addressed my comments and No major points remaining. However two minor points : 1) Auranofin is a Thioredoxin Reductase Inhibitor and not a Thioredoxin inhibitor, please revised throughout. This is important.

2) Authors should check references again: this is from their response to reviewers: "We cite the above works (27693380 and 28478025) in the Introduction (lines 119), in the Results section (Lines 304, 318)". the PMID numbers do not match the cited refs.

Reviewer #2: All of our concerns have been addressed. The revised manuscript makes a very nice contribution to understanding the factors that determine PIKfyve sensitivity in mammalian cells.

Reviewer #3: I've critically read the revised manuscript, including the strategies used by the authors to accommodate to the critiques. New data are added and some aspects, on the reviewers' advices, more carefully stated and described. The manuscript overall further improved and consists now over an elaborate set of data (+suppl) to support the conclusions on the role of ROS in PIKfyve inhibited cells. At least this reviewer has no further comments and is satisfied about the revision.

7. PLOS authors have the option to publish the peer review history of their article (what does this mean?). If published, this will include your full peer review and any attached files.

Reviewer #1: No

Reviewer #2: No

Reviewer #3: No

---

## [Author Response · Author response to Decision Letter 1]

13 Oct 2021

Response to Reviewers

We thank the reviewers for the additional comments and opportunity to further strengthen our investigative work. The detailed rebuttal and inventory of changes done to the manuscript are recorded below.

Reviewer #1: 

R1.1: The reviewers have addressed my comments and No major points remaining. However two minor points: 1) Auranofin is a Thioredoxin Reductase Inhibitor and not a Thioredoxin inhibitor, please revised throughout. This is important.

RESPONSE: We really appreciate the Reviewer catching this error on our part. We have corrected this in all occasions that we refer to auranofin as an inhibitor – it is an inhibitor of thioredoxin reductase.

R1.2: Authors should check references again: this is from their response to reviewers: "We cite the above works (27693380 and 28478025) in the Introduction (lines 119), in the Results section (Lines 304, 318)". the PMID numbers do not match the cited refs.

RESPONSE: Once again, we thank the Reviewer for catching this error. Our mistake was that we did not refresh the bibliography and so the old bibliography list was included. This is now corrected. Ref. 38 and 39 are PMID 27693380 and 28478025.

---

## [Decision Letter · Decision Letter 2]

18 Oct 2021

Reactive oxygen species prevent lysosome coalescence during PIKfyve inhibition

PONE-D-21-19956R2

Dear Dr. Botelho,

We’re pleased to inform you that your manuscript has been judged scientifically suitable for publication and will be formally accepted for publication once it meets all outstanding technical requirements.

Kind regards,

David Chau

Academic Editor

PLOS ONE

Additional Editor Comments (optional):

Reviewers' comments:

Reviewer's Responses to Questions

**Comments to the Author**

1. If the authors have adequately addressed your comments raised in a previous round of review and you feel that this manuscript is now acceptable for publication, you may indicate that here to bypass the “Comments to the Author” section, enter your conflict of interest statement in the “Confidential to Editor” section, and submit your "Accept" recommendation.

Reviewer #1: All comments have been addressed

2. Is the manuscript technically sound, and do the data support the conclusions?

Reviewer #1: Yes

3. Has the statistical analysis been performed appropriately and rigorously? 

Reviewer #1: Yes

4. Have the authors made all data underlying the findings in their manuscript fully available?

Reviewer #1: Yes

5. Is the manuscript presented in an intelligible fashion and written in standard English?

Reviewer #1: Yes

6. Review Comments to the Author

Reviewer #1: The authors have addressed all my comments and should be published as it is. Thank you for this detailed work.

7. PLOS authors have the option to publish the peer review history of their article (what does this mean?). If published, this will include your full peer review and any attached files.

Reviewer #1: No

---

## [Editor Report · Acceptance letter]

15 Nov 2021

PONE-D-21-19956R2 

Reactive oxygen species prevent lysosome coalescence during PIKfyve inhibition 

Dear Dr. Botelho:

I'm pleased to inform you that your manuscript has been deemed suitable for publication in PLOS ONE. Congratulations! Your manuscript is now with our production department. 

Kind regards, 

on behalf of

Dr. David Chau 

Academic Editor

PLOS ONE